# Local vs distributed representations: What is the right basis for interpretability?

## Abstract

Much of the research on the interpretability of deep neural networks has focused on studying the visual features that maximally activate individual neurons. However, recent work has cast doubts on the usefulness of such local representations for understanding the behavior of deep neural networks because individual neurons tend to respond to multiple unrelated visual patterns, a phenomenon referred to as "superposition". A promising alternative to disentangle these complex patterns is learning sparsely distributed vector representations from entire network layers, as the resulting basis vectors seemingly encode single identifiable visual patterns consistently. Thus, one would expect the resulting code to align better with human-perceivable visual patterns, but supporting evidence remains, at best, anecdotal. To fill this gap, we conducted three large-scale psychophysics experiments collected from a pool of 560 participants. Our findings provide (*i*) strong evidence that features obtained from sparse distributed representations are easier to interpret by human observers and (*ii*) that this effect is more pronounced in the deepest layers of a neural network. Complementary analyses also reveal that (*iii*) features derived from sparse distributed representations contribute more to the model´s decision.

Overall, our results highlight that distributed representations constitute a superior basis for interpretability, underscoring a need for the field to move beyond the interpretation of local neural codes in favor of sparsely distributed ones.

## 1 Introduction

One of the goals of explainable AI (XAI) in computer vision is to identify the visual features and characterize the representations used by deep neural networks (DNNs) to categorize images (Ribeiro et al., 2016; Sundararajan et al., 2017; Smilkov et al., 2017; Petsiuk et al., 2018; Selvaraju et al., 2017; Linsley et al., 2019; Fel et al., 2021; 2023c;a; Novello et al., 2022; Zhou et al., 2016; Bau et al., 2017; Cammarata et al., 2020b; Kim et al., 2018; Ghorbani et al., 2019). In general, identifying these features requires uncovering the visual patterns that drive the activation of units within a network.

This goal is shared with the study of biological vision, where there is extensive research over the last several decades focused on identifying the "preferred stimulus" of individual neurons in the visual cortex (Hubel & Wiesel, 1959; Lettvin et al., 1959; Tsunoda et al., 2001; Wang et al., 1996; Pasupathy & Connor, 2001; Quiroga, 2005). This approach to visual neuroscience reflected the dominant theory at the time, known as the "grandmother (cell)" theory , which postulates that information in the visual system is stored **locally** – at the level of single neurons – and that the visual system contains specific neurons that respond to particular objects or people (including one's own grandmother). Early XAI advances inspired by neuroscience research similarly focused on understanding local representations (Zhou et al., 2016; Bau et al., 2017). This led to the development of more sophisticated optimization methods to synthesize maximally activating images for individual neurons (Erhan et al., 2009; Zeiler & Fergus, 2014a; Yosinski et al., 2015; Olah et al., 2017; 2020; Nguyen et al., 2016a;b; Cammarata et al., 2020b).

This parallelism between XAI and neuroscience extends beyond vision as recent work has found neurons in multi-modal systems that respond to very high-level concepts beyond simple image appearance, including hand-drawing and text (Goh et al., 2021). Interestingly, the authors identified a "Halle Berry" neuron in CLIP, reminiscent of the neuroscience finding reported two decades ago in the human brain (Quiroga, 2005).

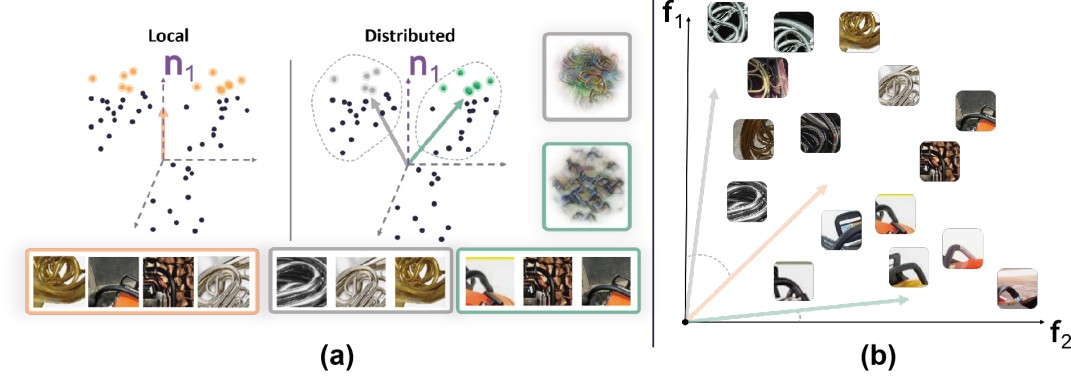

**(a)**               **(b)**

Figure 1: **(a)** • **Local** (neuron) versus • **Distributed** (sparsely distributed vector) visual representations. The activation of individual neurons may be driven by multiple unrelated visual elements (depicted in the images at the bottom) whereas distributed representations, obtained via dictionary learning methods, break down complex patterns into simpler ones corresponding to single visual features. **(b)** In practice, dictionary learning methods "disentangle" local activations to yield a new vector basis whose activation is driven by single features. The hope for interpretability is that those features align better with the set of features that humans can interpret $S = \{f_1, f_2, ..., f_n\}$.

At the same time, a paradigm shift is taking place in neuroscience, where the study of neural populations is quickly superseding the study of single neurons (for a review see Ebitz & Hayden (2021)) because the neural code is believed to be sparse and **distributed** rather than local (Haxby et al., 2001; Quiroga et al., 2008; 2013). Interestingly, a similar shift is emerging in XAI, because local representations are known to suffer from the "superposition" problem (Arora et al., 2018; Cheung et al., 2019; Olah et al., 2020; Elhage et al., 2022; Fel et al., 2023b): the number of features captured by DNNs might be larger than the number of neurons. Therefore, the neurons' activations might be driven by multiple unrelated features. To address this challenge, the XAI community has started using dictionary learning methods (Fel et al., 2023c;b; Bricken et al., 2023; Templeton et al., 2024) to project the activations of DNNs onto a new basis of vectors, each activated by a single distinct feature. From an interpretability perspective, representations driven by single features are more desirable because they are expected to be easier to understand by a human, *i.e.*, a unit responding to a single visual pattern –compared to multiple patterns– is inherently easier to interpret. An implicit hypothesis is that applying dictionary learning methods to network activations helps break down complex visual patterns into simpler ones corresponding to single features (Fig. 1.a), which, in turn, might be easier for humans to interpret (Fig. 1.b). Scaling up standard interpretability evaluations to study representations poses significant challenges (Colin et al., 2022). A practical alternative is to assess the ambiguity—or perplexity—of the visual features derived from these representations (Borowski et al., 2021) (see 3.2.1 for a more thorough elaboration on this point).

Despite the growing consensus that distributed representations constitute a stronger basis for interpretability compared to local ones (single neurons), empirical evidence remains scarce. This paper aims to fill this gap. Specifically, we contend that a representation can be considered superior if the features derived from it are more intelligible, *i.e.*, easier for humans to make sense of while being demonstrably used by the model in its decision-making process. By means of computational and psychophysics experiments, we set out to find which of the local vs. distributed representations better meets these two conditions.

In sum, the main contributions of this paper are as follows:

- We conduct three large-scale psychophysics experiments for a total of 15,720 responses from a pool of 560 participants, to evaluate the visual ambiguity of the features derived from local vs distributed representations (see Fig 1). In the process, we identify a potential semantic bias in the experimental protocols commonly used in the field (Borowski et al., 2021; Zimmermann et al., 2023), and provide an approach to at least partially mitigate it.
- Our findings provide strong evidence that (***i***) features derived from distributed representations are significantly easier for humans to interpret than features derived from local

representations, and (***ii***) this effect is even more pronounced in the deepest layers of a neural network.

- Additionally, we observe that (***iii***) models rely significantly more on features derived from distributed representations compared to those derived from local representations. Overall, our results suggest that distributed representations provide a substantially better foundation for the interpretability of models than local representations.

## 2  RELATED WORK

**From local to distributed representations in XAI.**    Early research on explainable AI (XAI) in computer vision developed attribution methods (Zeiler et al., 2011; Zeiler & Fergus, 2014a;b; Sundararajan et al., 2017; Smilkov et al., 2017; Fong & Vedaldi, 2017; Ancona et al., 2018; Shrikumar et al., 2017; Chattopadhay et al., 2018; Fel et al., 2021; Novello et al., 2022) to understand specific model predictions. These methods predominantly aimed to identify *"where"* are the most important pixels of an image, given a specific model prediction. Unfortunately, these methods fell short in explaining the *"what"* (Kim et al., 2018; 2022; Taesiri et al., 2022; Colin et al., 2022), *i.e.*, the visual features that the models rely on to make their predictions. As a result, new XAI methods, including Feature Visualization approaches (Nguyen et al., 2016b; Olah et al., 2017), were developed to provide insights into the features that neurons or model layers respond to. These efforts revealed that the neurons' activations can be driven by visually distinct features (Nguyen et al., 2016b; 2019; Cammarata et al., 2020a; Bricken et al., 2023). A similar behavior was observed in other fields, and notably in Natural Language Processing (NLP) (Elhage et al., 2022).

The neurons' tendency to respond to multiple unrelated visual elements indicates that single neurons might not align well with the model's internal representations. This phenomenon, known as super-position (Arora et al., 2018; Cheung et al., 2019; Olah et al., 2020; Elhage et al., 2022; Fel et al., 2023b) or feature collapse (Fel et al., 2023c), suggests that there could be significantly more features than neurons in a model. Consequently, interpreting neurons might be no more meaningful than interpreting arbitrary directions in the feature space. To achieve effective model interpretation, it is, therefore, essential to identify an interpretable basis that facilitates the extraction of meaningful features. This observation has led to increased interest, over the past five years, towards examining deep learning models by considering the distributed nature of latent space representations. More specifically, it spurred the development of concept extraction methods (Ghorbani et al., 2019; Zhang et al., 2021; Fel et al., 2023c;b; Graziani et al., 2023; Vielhaben et al., 2023) that leverage dictionary learning, especially overcomplete dictionaries. In NLP, sparse autoencoders (SAEs) (Elhage et al., 2022; Bricken et al., 2023; Cunningham et al., 2023; Tamkin et al., 2023) are extensively studied and regarded as a promising direction for discovering interpretable bases. In this paper, we are interested in comparing the suitability of local vs distributed representations to serve as a basis for the interpretability of deep learning models in computer vision.

**Human-evaluation of interpretability.**    Since the ultimate goal of XAI is to make complex models understandable to humans, it is essential to incorporate human evaluation in measuring interpretability through psychophysics experiments. Human evaluation serves as a critical benchmark for assessing whether the explanations provided by XAI systems are comprehensible, useful, and actionable for end-users. This human-centric approach ensures that interpretability methods are not just theoretically sound but also practically effective in real-world applications.

To the best of our knowledge, Borowski et al. (2021) were the first to quantify the interpretability of features through psychophysics experiments. Their approach involved visualizing unit responses by contrasting maximally and minimally activating stimuli. Specifically, Borowski et al. (2021) focused on studying features from local representations (i.e., single unit activations) of an Inception V1 (Szegedy et al., 2015) trained on ImageNet (Deng et al., 2009; Russakovsky et al., 2015). In their experiments, participants were shown maximally and minimally activating images from the ImageNet ILSVRC 2012 validation set (Russakovsky et al., 2015) to illustrate a specific feature. They were then asked to select which of two query images also activated the feature (see Fig. 2 for an example trial). As a proxy for interpretability, they measured the visual coherence of maximally activating stimuli of a given feature. It is maximal when the visual feature is unambiguous. The main conclusion from these experiments is that, from a human-centric perspective, maximally activating natural images are more effective for studying features than synthetically generated feature visualizations (Olah

et al., 2017). Zimmermann et al. (2021) proposed a variation of this task to investigate if humans gained causal insights from those visualizations. They tested the participants' ability to predict the effect of an intervention, such as occluding a patch of the image, on the activation of the unit. Both by means of a large-scale crowdsourced psychophysics experiment and measurements with experts, they found that synthetic feature visualizations (Olah et al., 2017) helped humans perform the task successfully. However, these visualizations did not provide a significant advantage over other visualizations, such as exemplary images. Finally, Zimmermann et al. (2023) extended the work of Borowski et al. (2021) to a broader set of deep learning architectures used for computer vision tasks, including a ResNet50 (He et al., 2016). Their main conclusion is that increasing the scale of the model does enhance the interpretability of the features. While both our work and that of Zimmermann et al. (2023) build upon the experimental protocol of Borowski et al. (2021), they focus on scaling Borowski et al. (2021)'s insights, whereas we adapt the protocol to generate insights into a different and novel research question: comparing the suitability of features derived from local vs distributed representations for interpretability.

Interestingly, in a similar vein but within the context of NLP, (Bricken et al., 2023) compare the interpretability of 162 features extracted from both local and distributed representations. One of the authors scored each feature using samples drawn uniformly across the spectrum of activation. Their findings indicate that features obtained from distributed representations are substantially more interpretable than those from single neurons. The research presented in this paper differs from this work in the application domain, the experimental design, single participant in their study vs 15,720 responses from a pool of 560 participants in our study.

## 3 METHODOLOGY

In this section, we first provide an overview of the technical methods used to create the conditions for the three psychophysics experiments described below.

### 3.1 TECHNICAL METHODS

**Model** All the experiments described in this paper were performed on a ResNet50 (He et al., 2016) from the Torchvision (Marcel & Rodriguez, 2010) library, pre-trained on ImageNet-1k (Deng et al., 2009).

**Sparse Distributed Representations** To compute sparse distributed representations for the distributed condition in our psychophysics experiments (see Section 3.2), we employed CRAFT (Fel et al., 2023c), an off-the-shelf dictionary learning method. Specifically, given a model $f : \mathcal{X} \to \mathcal{A}$ that maps from an input space $\mathcal{X} \subseteq \mathbb{R}^d$ to an activation space $\mathcal{A} \subseteq \mathbb{R}^p$ (e.g., any layer of the network), we compute the activations $\mathbf{A} = f(\mathbf{X}) \in \mathbb{R}^{n \times p}$, where $\mathbf{X} = [x_1, \ldots, x_n]^\top \in \mathbb{R}^{n \times d}$ represents a set of $n$ input data points. Each row $a_i \geq \mathbf{0}$ of $\mathbf{A}$ contains the non-negative activations for a given data point $x_i$, due to the use of ReLU activations. To obtain the sparse distributed representations, Non-negative Matrix Factorization (NMF) is applied to approximate $\mathbf{A}$ as:

$$(\mathbf{Z}^\star, \mathbf{D}^\star) = \operatorname*{arg\,min}_{\mathbf{Z} \geq \mathbf{0},\ \mathbf{D} \geq \mathbf{0}} \left\| \mathbf{A} - \mathbf{Z}\mathbf{D}^\top \right\|_F ,$$

where $\|\cdot\|_F$ denotes the Frobenius norm. Here, $\mathbf{Z} \in \mathbb{R}^{n \times k}$ are the **codes** (or concept coefficients), and $\mathbf{D} \in \mathbb{R}^{p \times k}$ forms the **dictionary** (or concept bank). Both $\mathbf{Z}$ and $\mathbf{D}$ are constrained to have non-negative entries and tend to be sparse due to the properties of NMF. The dictionary matrix $\mathbf{D}$ provides a new set of basis vectors (concepts) aligned with the activation patterns of the neural network, while $\mathbf{Z}$ contains the coefficients representing the original activations $\mathbf{A}$ in terms of these concepts. CRAFT is particularly well suited for our purposes as it has been shown to extract interpretable features from deep neural networks (Fel et al., 2023c).

**Feature importance** For an image $x_i$, let $\mathbf{v}_c$ be the vector it activates most strongly from an intermediate representation—it can be a sparse distributed vector in $\mathbf{D}$ or a neuron in $\mathcal{A}$—, and let $y_i$ be the logit score for $x_i$. To assess $\mathbf{v}_c$'s importance, we perform an ablation by setting the activation along this dimension to zero for $x_i$, resulting in modified activations $\mathbf{a}_i'$. These modified

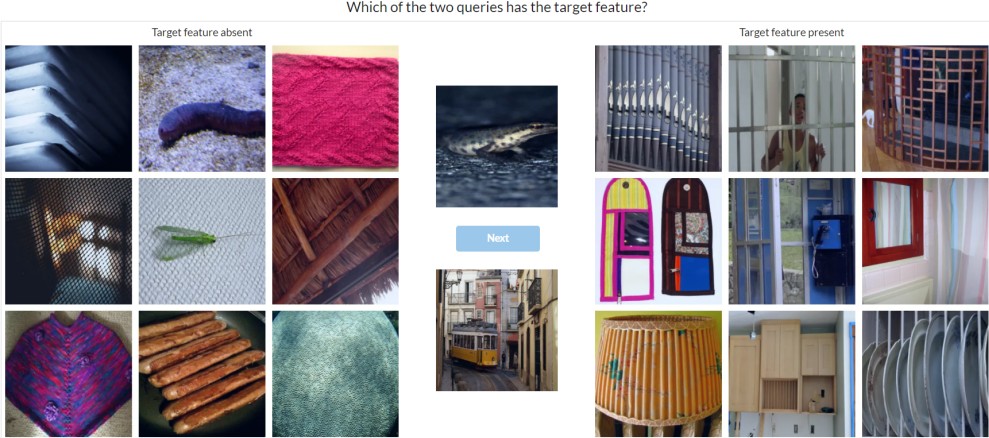

Figure 2: **Illustration of a trial.** Example of a trial in our study corresponding to Experiment I, distributed representation condition of a unit located in $layer2.0.bn2$. Two panels of 9 reference images are located on the left and right-hand side of the display, separated by 2 query images in the center. Participants were asked to select the query image they believed shared the same visual elements as the reference images displayed on the right panel, corresponding to maximally activating stimuli. The less ambiguous this shared visual element is—the more visually coherent the set of images—the more likely participants are to select the correct query. In this case, the correct query is the bottom image depicting a yellow tram.

activations are then propagated to the output layer to compute a new logit score, $\boldsymbol{y}_i'$. One can assess the importance of $\mathbf{v}_c$ by measuring as the difference in logit scores:

$$\Delta \boldsymbol{y}_i = \boldsymbol{y}_i - \boldsymbol{y}_i' \tag{1}$$

The more important the feature driving the activation of $\mathbf{v}_c$ is for the model, the higher the drop in logit score.

### 3.2 PSYCHOPHYSICS STUDIES

#### 3.2.1 EXPERIMENTAL PROTOCOL

Interpretability is a human-centric attribute. Hence, to shed light on how well local vs distributed representations serve as a basis for the interpretability of deep neural networks, we performed three large-scale online psychophysics experiments. The three experiments focus on evaluating the interpretability of features extracted from both types of representations. In practice, given the challenges of scaling up standard interpretability evaluations to study representations (Colin et al., 2022), we adapt the experimental protocol of Borowski et al. (2021) to measure the ambiguity, or perplexity, of the visual feature as a proxy for its interpretability. More precisely, given a feature and a set of images that illustrate it—*e.g.*, maximally activating images—, this protocol measures how visually coherent humans find this set of images. The more visually coherent it is, the more likely they are to correctly identify another member of this set, namely the correct query image. Hence, in our results, we report the visual coherence of a feature as the proportion of participants that correctly identified the query image. This visual coherence should be maximal when the feature represents a single visual pattern that people can interpret.

Each participant was assigned to one of two conditions: local or distributed representation. After successfully finishing a practice session composed of 9 trials, participants were asked to sequentially perform 40 different trials of the same task, where each trial corresponded to a different network unit. For each selected network unit and trial, participants were shown on the left and right-hand side of the screen two panels of 9 reference images each, separated by 2 images in the center, which were denoted as *queries*, as illustrated in Figure 2. The task across all experiments consisted of selecting the query image that participants believed shared the same visual elements as the 9 reference images

displayed on the right panel. The selection of the reference images shown in each panel differed depending on the experiment, as explained below.

### 3.2.2 NEURON AND STIMULI SELECTION

**Neuron selection** In line with the literature (Zimmermann et al., 2023), we aimed to obtain a representative sample of units within the different layers of the ResNet50. As a starting point, we selected the 80 units reported in (Zimmermann et al., 2023) for a ResNet50 (He et al., 2016) model. Given that CRAFT requires positive activations, we replaced the neurons from the convolutional layers (which could have non-positive activations) to their equivalent in the following batchnorm layer ($layer1.1.conv1$ neuron 53 -> $layer1.1.bn1$ neuron 53). As a result, we selected 80 neurons distributed across 43 layers that are grouped in four blocks. Thus, the names of the layers used in this paper ($layer1$ through $layer4$) are directly obtained from the modules of the ResNet50 Pytorch implementation.

**Stimuli selection for the local representation condition** For each of the selected neurons, we identified $2,900$ images from the validation set of ImageNet ILSVRC 2012 (Russakovsky et al., 2015) obtained as follows: the $2,500$ most strongly activating images and the $400$ least strongly activating images. Following the procedure described by Borowski et al. (2021), we illustrated a visual feature through both maximally activating (images that possess the feature, see Fig. 2 right panel) and minimally activating (images that do not possess the feature, see Fig. 2 left panel) stimuli. Again, like in previous work (Borowski et al., 2021), for the former, we selected a random sample of 9 images from the top 150 images; for the latter, we uniformly sampled 9 images from the bottom 20 images. We created 10 different trials per feature following this procedure to ensure image independence in the results. When participants were assigned to a feature, we randomly sampled one of those 10 trials to illustrate the feature.

**Stimuli selection for the distributed representation condition** Our hypothesis is that applying dictionary learning methods to local representation allows us to disentangle the superposition of features that might drive the activation of a neuron. Hence, to test this hypothesis, all the distributed representations studied in this work are obtained from the local representation. In practice, for each neuron in the local condition, we selected the top 300 maximally activating images, *i.e.*, those that most strongly led to the activation of the neuron. We used CRAFT to identify a new basis of distributed vectors through which their activations can be expressed. Following the recommended procedure in (Fel et al., 2023c), we constrained the dimensionality of this new basis to $d = 10$. From these 10 directions, we selected the one that was the most frequently the most activated across the 300 images. Once a direction was chosen, we ranked the $2,900$ images according to their activation in that direction to obtain the images that illustrate a visual feature in the distributed representation.

### 3.2.3 EXPERIMENTS

In this section, we present the details of each of the three psychophysics experiments conducted in this work.

**Experiment I** This experiment is an adaption of the methodology proposed by (Borowski et al., 2021) with two conditions: local vs distributed representations. An illustration of a trial belonging to Experiment I can be found in Figure 2. In this experiment, we performed a total of 1,600 trials (80 neurons × 2 conditions × 10 trials per unit).

**Experiment II** The main objective of Experiment II was to control for potential semantic confounding variables present in Experiment I. If the reference images that possess/do not possess the feature of interest belong to very different semantic groups (classes), then it could be possible to solve the task through simple semantic grouping. Figure 3 illustrates this phenomenon: in the example depicted in the figure, it is easier to solve the trial by inferring that the feature of interest is not about a monkey than it is to infer the actual visual feature present in the reference images. The goal of Experiment II is to control for this potential semantic confound as follows: given a set of reference images that possess the feature of interest, we extract their semantic labels from ImageNet and aim to find within the 400 minimally activating images a set of 10 images that share the same distribution of semantic labels. We define four levels of semantic similarity ($level\ 0$ to $level\ 3$). In $level\ 0$, the

Figure 3: **Illustration of the role of semantics**. Example of a trial from Experiment I in the local representation condition. In this case, the task can be trivially solved by simply relying on semantics. By observation of the minimally activating stimuli (left panel), it is easy to conclude that the neuron of interest is not a monkey detector, yet, it is hard to articulate what is the visual feature captured by the neuron (images in the right panel).

labels from ImageNet are used to determine the semantic similarity, whereas in $levels$ 1 through 3 we obtain the labels by moving up one branch in the WordNet (Fellbaum, 2010) hierarchy. In practice, we performed an iterative search for each trial starting from $level 0$. If there are not 10 images in the 400 minimally activating set that share labels with the reference images at a given level of semantics, the process is restarted, this time using labels from the next and broader level of semantics. We continued this process until 10 images from the minimally activating set shared the distribution of semantic labels with the reference images. In cases where it was impossible to identify 10 images, the feature was excluded from the experiment. Only one feature was fully excluded for both the local and distributed conditions due to this factor.

**Experiment III**    Finally, recognizing that the adopted experimental protocol was originally designed to assess whether features are better understood using natural or synthetic visualizations, we devised Experiment III to explore the impact of combining natural images with feature visualizations (See more details in Section D).

### 3.2.4    PARTICIPANTS

A total of 560 participants were recruited to take part in Experiments I, II, and III through the online platform Prolific [1]. All participants were native English speakers who reported not being visually impaired and completed the study on a laptop or desktop computer (not a mobile phone). They provided informed consent electronically and were compensated $2.75 for their time ($\sim 10-13$ min). The protocol was approved by the University IRB. Based on the power analysis of (Zimmermann et al., 2023), a minimum of 60 participants per condition (120 participants per Experiment) was needed to obtain statistically robust results. Furthermore, participants were required to (1) succeed in at least 5 of the 9 practice or instruction trials and (2) correctly answer at least 4 of the 5 attentiveness tests (catch trials) that were randomly inserted in the experiment. As a result, we analyzed the data corresponding to 138, 133, and 122 participants from Experiments I through III, respectively.

---

[1] www.prolific.com

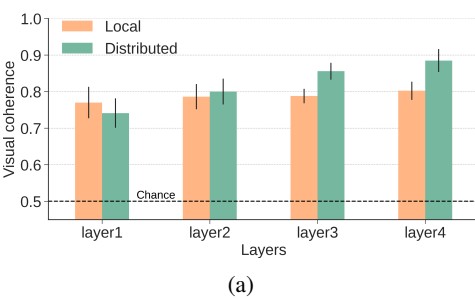 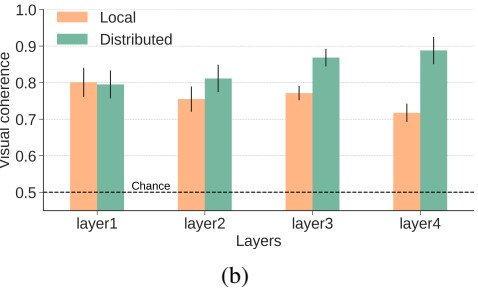

(a)                                                              (b)

Figure 4: **Per-layer results for Experiment I (a) and II (b)**. Given a feature and a set of images to illustrate it, we assess how visually coherent participants find this set of images—or how unambiguous the feature is. More precisely, we measure the proportion of participants that are able to identify the query image which is also part of this set of images. In both experiments, a clear trend emerges where features appear significantly less ambiguous in the distributed representation than in the local representation condition, particularly in the deeper layers of the network.

## 4 RESULTS

In this section, we summarize the results obtained from analyzing the responses from the 138, 133, and 122 participants who successfully completed Experiments I, II, and III, as well as the results from our feature importance analysis.

Unless stated otherwise, all of our behavioral data analyses consist of generalized logistic mixed-effects regressions (GLMER), with trial accuracy (1 vs. 0) as the dependent variable. The random-effects structure included both a by-participant and a by-unit [2] random intercept, as well as a by-unit random slope for the `condition` variable. We used the lme4 (Bates et al., 2015) package in R (R Core Team, 2021) to fit the models and lmerTest to obtain p-values for the fixed effects, with an $\alpha$-level of 0.05 for statistical significance.

**Reproduction of previous results.** Given that the experimental protocol used in Experiment I, with the local representation condition, is the same as the one described in (Zimmermann et al., 2023), we first evaluate to which degree our results corroborate previous research. For the ResNet50, Zimmermann et al. (Zimmermann et al., 2023) report an average task performance of $83.0\% \pm 2.0$ [3]. In our experiment, we obtain an average performance of $78.8\% \pm 1.5$. Given the results' similarity and that the selected units are not exactly the same (as described in Section 3), we conclude that Experiment I reproduces previously reported findings regarding local representations. This result also serves as an external validation of our experimental protocol.

**Human performance is superior in the distributed representation condition.** Based on our main hypothesis that a distributed representation constitutes a better basis for interpretability than a local representation, we predicted that participants would be better at selecting the maximally activating query image in the distributed condition. In Experiment I, the average performance across participants in the distributed condition was $83.5\% \pm 1.4$ [4], when compared to $78.8\% \pm 1.5$ in the local condition. A GLMER with `condition` as a predictor revealed a statistically significant disadvantage for the local condition: $\beta_{condition} = -0.47, SE = 0.23, z = -2.04, p = 0.04$. This result was corroborated both in Experiments II (see examples of trials in Fig 8, 9, 10) and III, with a mean performance of participants in the distributed condition of $85.1\% \pm 1.4$ *vs.* $76.2\% \pm 1.6$ in the local condition, $\beta_{condition} = -0.93, SE = 0.24, z = -3.93, p = < .001$ and a mean performance of $80.0\% \pm 1.6$ (distributed) *vs.* $74.1\% \pm 1.7$ (local), $\beta_{condition} = -0.42, SE = 0.17, z = -2.47, p = 0.01$, respectively.

---

[2] Here, a unit refers to either a neuron in the local representation condition or to a specific direction of the dictionary in the distributed representation condition.

[3] Values inferred from Figure 3 in their paper.

[4] The values reported correspond to a 95% confidence interval.

**Semantic control matters, but not significantly.** Figures 4a and 4b depict the results obtained without (Experiment I) and with (Experiment II) a semantic control applied to the stimuli, respectively. At first glance, the results seem consistent in both scenarios, except for the results corresponding to neurons in layer 4 where the performance of participants in the local condition suffers a drastic decrease in Experiment II. This result is coherent with the intuitive observation that the model aggregates class-specific information the closer the layer is to the output layer. Hence, neurons located in deeper layers are more likely to respond highly to features that correspond to certain classes and not at all to features belonging to other classes, *i.e.*, the semantic confounds are expected to be higher for those neurons. To investigate further the role of controlling the semantics, we pooled the data from Experiments I and II and included both `experiment` and an interaction term `condition:experiment` as predictors in a GLMER. We found a significant main effect for `condition`, $\beta_{neuron} = -0.47, SE = 0.23, z = -2.04, p = 0.04$, but not for `experiment`. Interestingly, we did not obtain evidence for a significant interaction between `experiment` and `condition`, $\beta_{neuron:Exp2} = -0.46, SE = 0.30, z = -1.53, p = 0.13$. As it was not possible to semantically control all trials in Experiment II to the same extent, we also tested for an interaction between `condition` and `semantic control` instead of the variable `experiment`. `Semantic control` was coded as a categorical variable with 4 levels: no control, 1, 2, and 3, corresponding to the previously described levels in Section 3.2.3. We compared two GLMERs: one including only the main effects of `condition` and `semantic control`, and another also considering the interaction between these two variables. Based on the Akaike Information Criterion, the model without the interaction was preferred (9116 vs. 9120). In sum, while the nominal disadvantage of the neuron condition is larger in Experiment II (semantic control) than in Experiment I (no semantic control), none of the performed statistical tests yielded a significant difference between these two experiments. This result can be partially explained by the unequal distribution of trials across semantic levels (see fig 11) and by the quality of the semantic control achieved once getting to a certain level in the Word-Net hierarchy. We leave to future work a finer-grained study of the role of semantics in the interpretability of features.

**The deeper the layer, the more prominent the benefits of the distributed representation.** Figures 4a, 4b and 7 illustrate the per-layer results obtained in Experiments I, II and III, respectively. We find evidence to suggest that the benefits of the distributed representation increase with the depth of the layer from which we select the unit that the participants were interpreting. Indeed, we identified a significant main effect for `unit` and also for an interaction term `unit:condition`, *i.e.*, the advantage of the distributed condition increases as the units belong to deeper layers. This result is consistent across the 3 experiments: $\beta_{depth} = 0.08, SE = 0.02, z = 3.92, p < .001$ and $\beta_{depth:Exp1} = -0.06, SE = 0.03, z = -2.22, p = 0.03$ for Experiment I; $\beta_{depth} = 0.07, SE = 0.02, z = 3.27, p = 0.001$ and $\beta_{depth:Exp2} = -0.09, SE = 0.03, z = -3.6, p < .001$ for Experiment II; and $\beta_{depth} = 0.05, SE = 0.02, z = 2.6, p = 0.01$ and $\beta_{depth:Exp3} = -0.06, SE = 0.03, z = -2.59, p = 0.01$ for Experiment III.

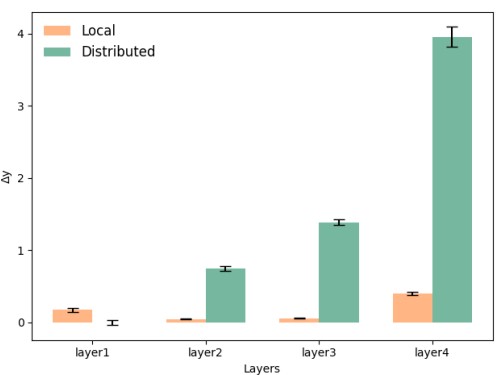

Figure 5: **Feature importance.** We measure the importance of a feature as the average drop in logit score $\Delta y$ for the 300 most activating images when the feature is occluded. Except for $layer1$, we find that the model relies significantly more on features derived from the distributed representation than on features from local representations, $z = -5.86, p < .001$ (Mann-Whitney U test).

**The model relies more on features derived from the distributed representation.** We evaluate the importance of every feature used in our psychophysics experiment by measuring the average drop in logit score for the 300 most activating images when the feature is occluded (see Eq. 1). Overall, we find that the model relies significantly more on features derived from distributed representations than from local representations: $z = -5.86, p < .001$ (Mann-Whitney U Test), the only exception

being $layer1$ (Fig 5). Interestingly, and in line with the rest of our results, the deeper the layer, the larger the difference between the local vs distributed representations.

## 5 DISCUSSION

**Conclusion** In this work, we have investigated and compared the suitability of local and distributed representations to serve as a basis for the interpretability of deep neural networks. We contend that the best-suited basis possesses a dictionary of features that is (a) more aligned with the set of features that human observers can interpret while (b) being demonstrably more important for decision-making by the model. Through psychophysics experiments, we consistently find that features derived from distributed representations are easier for humans to interpret, particularly when features are derived from deeper layers in the network. Equally important, we find that the model relies significantly more on those features to make its decisions. To the best of our knowledge, our results provide the first empirical evidence of the superiority of distributed representations over local ones for the interpretability of deep neural networks.

**Limitations** Our study is not exempt from limitations. First, the methodology proposed by Borowski et al. (2021) and followed by Zimmermann et al. (2023) utilizes the entire ImageNet validation set with the goal of studying a broad range of stimuli and thereby increasing the likelihood of identifying stimuli that are representative of neurons' selectivity. While such motivation is sound, it poses a significant challenge when performing psychophysics experiments: neurons that are selective to class-specific features (*e.g.*, fish scale) will be maximally activated when presented with stimuli that belong to the corresponding classes (*e.g.*, fish) and minimally activated when provided with stimuli that belong to other classes (*e.g.*, dogs). In those cases, the task can be solved trivially using semantics. The design of Experiment II reflects an initial attempt to mitigate this confounding factor. However, both quantitative results and a manual exploration of the trials by the authors hint at only a partial success of the adopted methodology to address this challenge. We leave to future research a further refinement of the experimental protocol.

Second, while the original work (Borowski et al., 2021) this manuscript builds on aimed at measuring the *interpretability* of features, we have been more conservative with the terminology employed in this paper as we believe that the current experimental protocol only measures a loose proxy of interpretability. Interpretability, or understanding, of a complex visual pattern usually requires procedural learning with feedback (Ashby & Maddox, 2005). Given that, in practice, this protocol measures a score based on a set of images presented in a single trial, it seems unreasonable to expect this score to capture interpretability. In contrast, it seems more reasonable to expect this score to characterize something more visual, like the ambiguity of the features, i.e., how complex it might be to interpret the features if one had to. Such ambiguity can be inferred, at least in part, by the visual coherence of the set of images. Zimmermann et al. have indeed shown that responses from participants can be accurately predicted based on visual coherency—or perceptual similarity—of a trial.

While we believe that these metrics are useful proxies and that features achieving a high score at them are desirable for interpretability, we want to emphasize that we do not claim that it means that they *are* interpretable. Perceptual similarity differs from interpretability; being able to group images correctly does not necessarily imply a deep understanding of the underlying factors driving those groupings or the ability to explain why the images belong together based on intrinsic features or causal relationships.

Nevertheless, we believe that the adopted experimental protocol serves as a solid foundation for developing a scalable and robust paradigm to measure the interpretability of features in deep learning models. We leave to future work the development of such a paradigm.

## ACKNOWLEDGEMENT

This work was funded by the ONR grant (N00014-24-1-2026), NSF grant ( IIS-2402875 and EAR-1925481) and the ANR-3IA Artificial and Natural Intelligence Toulouse Institute (ANR-19-PI3A-0004). The computing hardware was supported in part by NIH Office of the Director grant #S10OD025181 via the Center for Computation and Visualization (CCV) at Brown University. J.C. and N.O. have been partially supported by Intel Corporation and funding from the Valencian Government (Conselleria d'Innovació, Industria, Comercio y Turismo, Direccion General para el Avance de la Sociedad Digital) by virtue of 2022-2023 grant agreements (convenios singulares 2022, 2023). J.C. has also been partially supported by a grant by Banco Sabadell Foundation.

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

# APPENDIX

## A  PRACTICE TRIALS

All participants were shown 9 practice trials to help them understand the core idea of the task. The images for these trials did not overlap with the images used in any of the experiments, preventing them from influencing our results. They were selected to represent specific features on the maximum-activating grid: checkerboard, veiny, green, round, blue, rough fur, yellow, straight lines, and magenta. Images on the left side of the grid in practice trials had no coherent pattern, and were randomly sampled without replacement from a set of images with features including whiskers, spikes, droplets, and liquid flow.

Practice trials were customized for each experiment, to keep all variables controlled. For the feature visualization psychophysics, practice trials were generated using the same methods as the experimental images. They were chosen from neurons in convolutional layers to show the same features that were included in the practice trials for other experiments.

## B  ATTENTIVENESS TESTS

Attentiveness tests were included at random points in the study. Following the methodology adopted in previous work (Borowski et al., 2021), they were based on a simple premise: if the query image is also present in the grid of maximum-activating images, participants paying attention should get near-perfect accuracy. We generated attentiveness tests for each experiment using the same procedure as was used for the experimental trials. To ensure that the attentiveness tests did not overlap with experimental trials, we chose maximum- and minimum-activating images from neurons from convolutional layers. Rather than sampling 10 maximum-activating images, we sampled 9 and put the query in a random position in the maximum-activating grid. The attentiveness tests were generated for all experiments using the same neurons from the model to guarantee consistency.

## C  IMAGE SECTION FOR THE DISTRIBUTED REPRESENTATION CONDITION.

We originally tested 2 ways to select the direction from the distributed representation obtained by CRAFT to be kept for the main experiment. The first one is the one explained in the main section of the paper, namely: from the 10 directions, we selected the direction that was the most often the most activating one across the 300 images. However, we considered a second alternative where we selected the direction that was the most often the most activating one across the 2900 images. We ran the psychophysics experiments on both alternatives. The average performance obtained on this alternative was of $81\% \pm 1.4$ and we find no effect of the condition variable between this distributed condition vs the local condition describe in the main paper: $\beta_{condition} = -0.34, SE = 0.25, z = -1.39, p = 0.164$. Further quantitative and qualitative analysis done on both alternatives have led us to keep the first alternative for the reminder of the experiments.

# D SUPPLEMENTARY INFORMATION ABOUT EXPERIMENT III

**Experimental protocol**   Given that the adopted experimental protocol was originally designed to evaluate the informativeness of feature visualizations, we devised Experiment III to shed light on the value that a state-of-the-art feature visualization method (MACO) would add to the task. We subscribe to the idea that feature visualizations are more suitable to complement natural images than to replace them. Thus, Experiment III implements a protocol similar to the *mixed* condition reported by Zimmermann et al. (Zimmermann et al., 2021). In practice, Experiment III was based on Experiment II, but in each trial, 4 of the natural images displayed both on the left and right panels were replaced by 4 feature visualizations, as depicted in Figure 6.

**Method**   To synthesize feature visualizations for Experiment III (see Section 3.2), we used MACO (Fel et al., 2023a), inspired by the method by Olah et al. (Olah et al., 2017). MACO generates feature visualizations with a natural Fourier amplitude spectrum by fixing the amplitude spectrum to the empirical mean derived from natural images. Specifically, MACO optimizes directly in the Fourier domain $\mathcal{F}(\boldsymbol{x})$ by modifying the phase $\boldsymbol{\varphi}$ of the target image while keeping the magnitude $\boldsymbol{r}$ of the spectrum fixed. This constraint on the magnitude helps prevent high-frequency artifacts and ensures that the resulting images remain visually coherent. Formally, let $\boldsymbol{r}$ denote the average amplitude spectrum computed across the ImageNet dataset, and let $\boldsymbol{d}$ be a target concept direction to be maximized. MACO solves the following optimization problem:

$$\boldsymbol{\varphi}^{\star} = \arg\max_{\boldsymbol{\varphi}} \left( \boldsymbol{f} \left( \mathcal{F}^{-1} \left( \boldsymbol{r} \circ e^{i\boldsymbol{\varphi}} \right) \right) \cdot \boldsymbol{d} \right), \quad \text{and} \quad \boldsymbol{x}^{\star} = \mathcal{F}^{-1} \left( \boldsymbol{r} \circ e^{i\boldsymbol{\varphi}^{\star}} \right).$$

where $\boldsymbol{x}^{\star}$ is the feature visualization obtained after optimization, $\mathcal{F}$ denotes 2-D Discrete Fourier Transform (DFT) on $\mathcal{X}$, $\mathcal{F}^{-1}$ its inverse and $\circ$ represents element-wise multiplication. Additionally, MACO introduces an attribution-based transparency mask to highlight spatially important regions in the visualizations, enhancing interpretability.

**The value of feature visualization remains unclear.**   Finally, Figure 7 depicts the results obtained from Experiment III, which combined natural images with synthetic images corresponding to feature visualizations using MACO. Similarly to (Zimmermann et al., 2021), we do not find that mixing stimuli from feature visualization with natural images helped participants perform better at the task. In fact, participants performed overall significantly worse in Experiment III than in Experiment II: $\beta_{depth:Exp3} = -0.63, SE = 0.19, z = -3.3, p < 0.001$. Interestingly, the worst performance was observed in the local condition for units located in the deepest layers of the network, with a drop in performance from 80.2% in Experiment I to 68.3% in Experiment III. Furthermore, the difference in the per-layer task performance obtained in layer 4 in each condition is 8.2%, 17.1%, and 18.9%, respectively. We leave to future work the investigation into the specific factors that contributed to this decline in performance. Potential avenues for future research include exploring the cognitive load imposed by mixed stimuli and identifying optimal conditions under which feature visualization might enhance rather than hinder task performance.

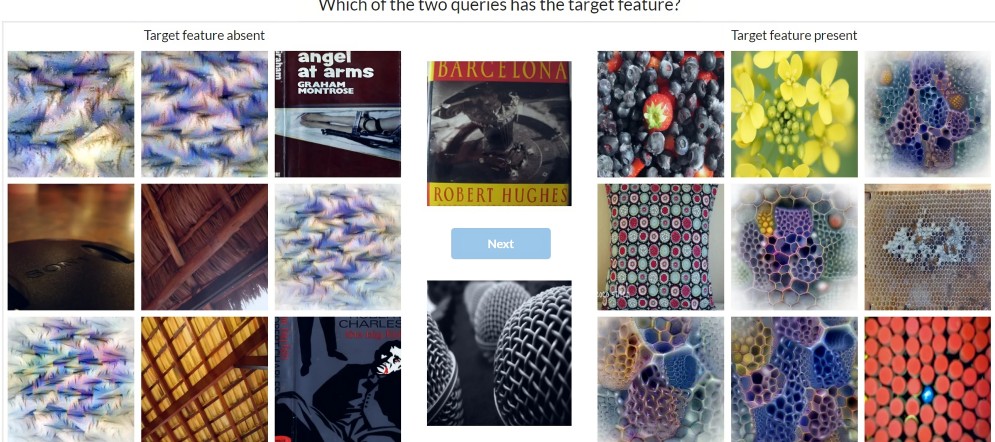

Figure 6: **Illustration of a trial from Experiment III**. Example of a trial from $layer4.2.bn3$ in the distributed representation condition from Experiment III. Note how the reference images on the left and right panels contain a mix of feature visualizations using MACO and natural images.

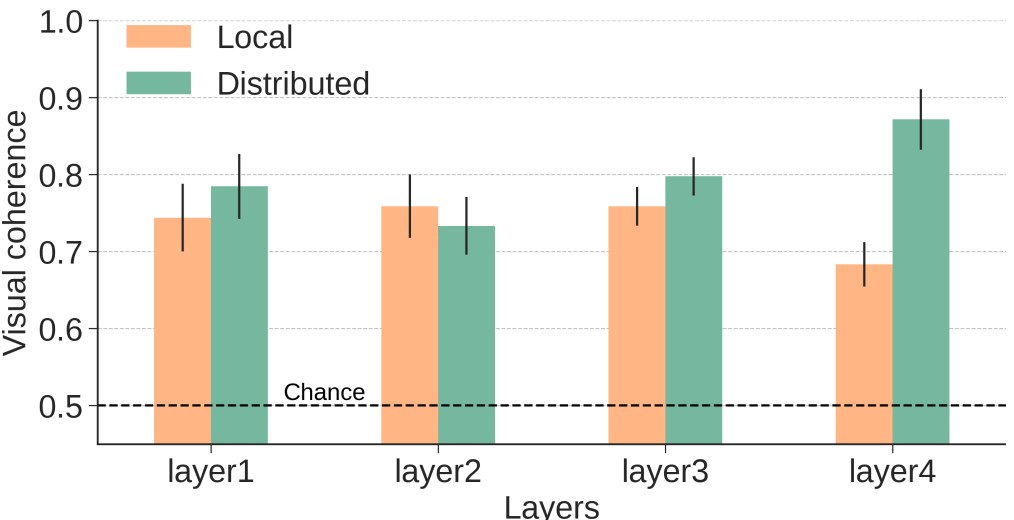

Figure 7: **Per-layer results for Experiment III**. Given a feature and a set of images to illustrate it, we assess how visually coherent participants find this set of images—or how unambiguous the feature is. More precisely, we measure the proportion of participants that are able to identify the query image which is also part of this set of images. Similarly to the results from experiment I and II, a clear trend emerges where features appear significantly less ambiguous in the distributed representation than in the local representation condition, particularly in the deeper layers of the network.

# E   FURTHER ILLUSTRATION OF LOCAL VS DISTRIBUTED TRIALS

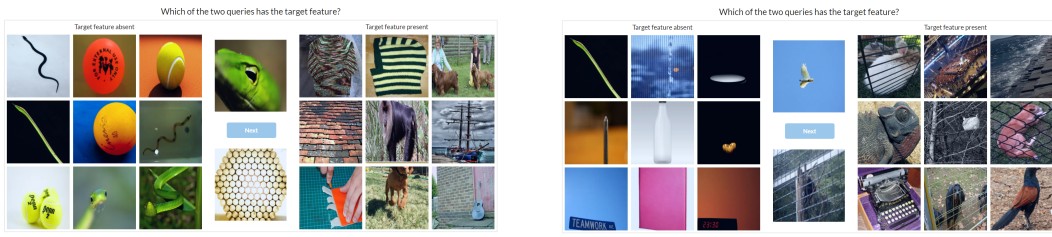

(a) Local representations    (b) Distributed representations

Figure 8: **Layer2.** This figure illustrates a trial used to assess the features encoded in layer2.0 either by the neuron 52 (a) or at least partially through the neuron 52 (b).

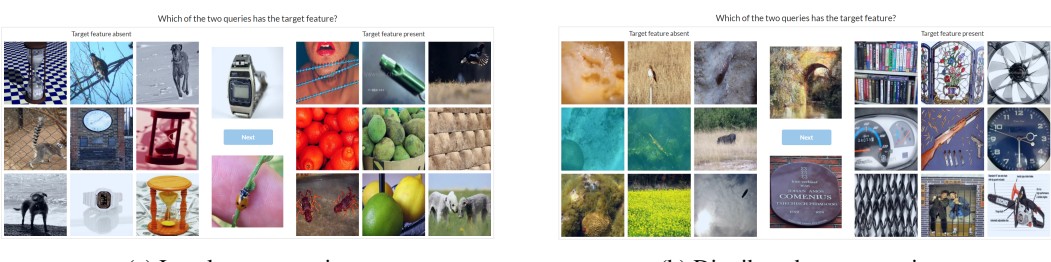

(a) Local representations    (b) Distributed representations

Figure 9: **Layer3.** This figure illustrates a trial used to assess the features encoded in layer3.1 either by the neuron 957 (a) or at least partially through the neuron 957 (b).

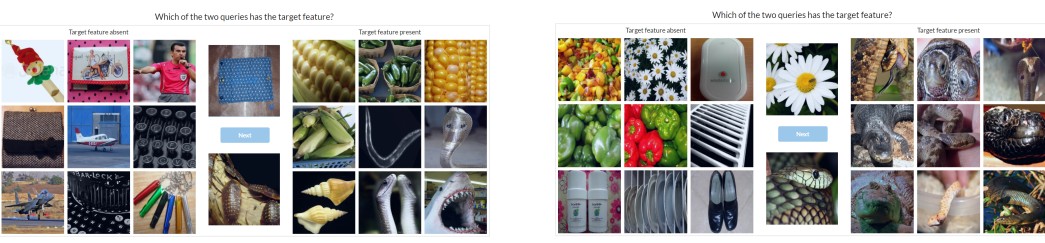

(a) Local representations    (b) Distributed representations

Figure 10: **Layer4.** This figure illustrates a trial used to assess the features encoded in layer4.2 either by the neuron 259 (a) or at least partially through the neuron 259 (b).

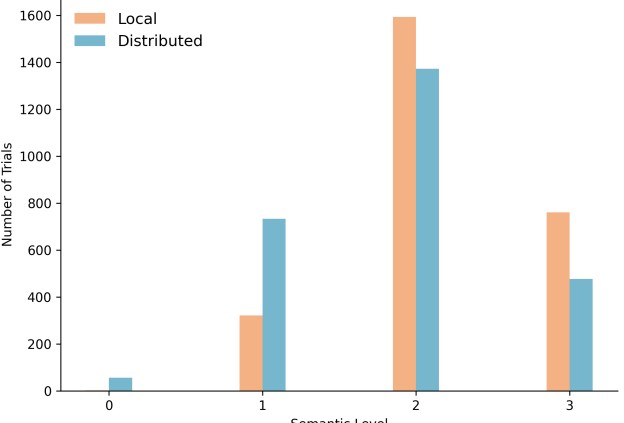

Figure 11: Number of trials per semantic control level, for both local and distributed representation.

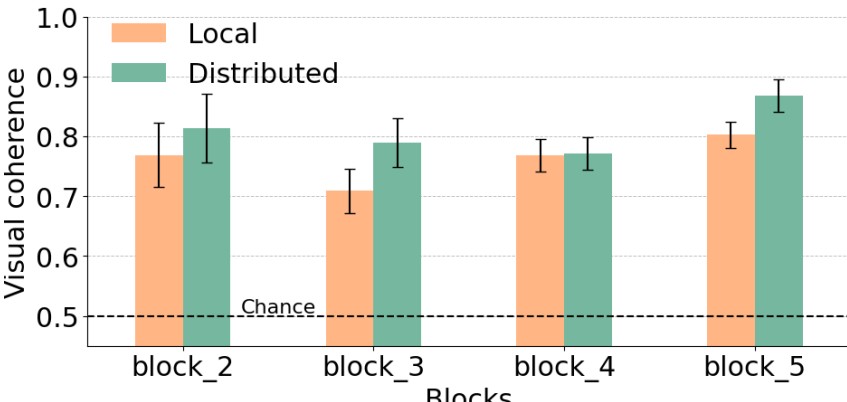

Figure 12: **Per-layer results for Experiment IV (VGG16)**. Given a feature and a set of images to illustrate it, we assess how visually coherent participants find this set of images—or how unambiguous the feature is. More precisely, we measure the proportion of participants that are able to identify the query image which is also part of this set of images. Similarly to the results from experiments I, II, and III, a trend emerges where features appear less ambiguous in the distributed representation than in the local representation condition. Nevertheless, the effect of depth appears more nuanced than in the ResNet50.

## F   LOCAL VS DISTRIBUTED: RESULTS FROM A VGG16

We extend the experiments to a VGG16 from the Torchvision (Marcel & Rodriguez, 2010) library, pre-trained on ImageNet-1k (Deng et al., 2009). We randomly select 80 neurons across the VGG16, and we follow the same methodology as previously for the selection of stimuli for the local and distributed conditions. For the psychophysics experiment, we follow the experimental protocol of *Experiment II* where we control for semantic confounds. We analyze the data corresponding to 132 participants from Prolific.

**Human performance is superior in the distributed representation condition.**   Based on our main hypothesis that a distributed representation constitutes a better basis for interpretability than a local representation, we predicted that participants would be better at selecting the maximally activating query image in the distributed condition. In Experiment IV, the average performance across participants in the distributed condition was $81.5\% \pm 1.5$, when compared to $77.1\% \pm 1.6$ in the local condition. A GLMER with `condition` as a predictor revealed a statistically significant disadvantage for the local condition: $\beta_{condition} = -0.45, SE = 0.22, z = -2.04, p = 0.04$.

**The benefit of the distributed representation does not increase systematically with depth.** Figures 12 illustrates the per-layer results obtained in Experiments IV (VGG16). While the superiority of distributed representation appears again here in the deeper layer, we do not see a systematic trend between depth and the performance of participants in our psychophysics experiments. This observation is reflected in our analysis as we do not find that the advantage of the distributed condition increases as the units belong to deeper layers: $\beta_{depth:Exp4} = -0.06, SE = 0.05, z = -1.27, p = 0.20$.

**Takeaway.**   In general, we expect deeper layers to contain more information, and we expect a model to rely more on superposition to encode information in those layers. Based on those assumptions, we expected participants to systematically benefit more from distributed representation when exposed to features from deeper layers. Yet, while our new results strengthen our claim that there appears to be no downside to studying features using distributed representation, they portray a more complex picture as to when they become necessary.

# G  Do features from *local* vs. *distributed* representations differ in obvious ways?

Following the useful suggestions of a reviewer we use existing metrics to probe if features from local vs. distributed representations differ in obvious ways.

## G.1  Complexity

**Kolmogorov complexity.**  The first hypothesis we test is whether features from distributed representation are easier to interpret than features from local conditions because they are less complex. While we do not have access to the feature per se, we have access to the images that possess the features as a proxy. While measuring the complexity of images is still an open research question, there are works that show that the Kolmogorov complexity of images (Li et al., 2004) correlate well with human-derived ratings for the complexity of natural images (Forsythe et al., 2008; Forsythe, 2009). In practice, we follow previous works (Li et al., 2004; de Rooij & Vitányi, 2006) and use a standard compression technique (JPEG) to approximate the Kolmogorov complexity of images. The hypothesis is that the more an image can be compressed, the less complex the features that compose the images.

**Methodology**  For a given target feature, we conducted T trials T ∈ [1, 10], based on how many trials can be semantically controlled). In each trial, we had access to 10 images that possess the target feature. We measured the average Kolmogorov complexity of these 10 images by recording their compressed file sizes after JPEG compression. We then averaged these complexities across all trials for each feature, resulting in a measure of Kolmogorov complexity for that feature. This methodology is applied to every feature.

**Results.**  We did not find a significant correlation between the Kolmogorov complexity of the images and the visual coherency scores from Experiments II (ResNet50) (Fig. 13 and IV (VGG16) (Fig 14). This suggests that while complexity might be a factor, the superiority of features from distributed representations cannot be explained by complexity alone.

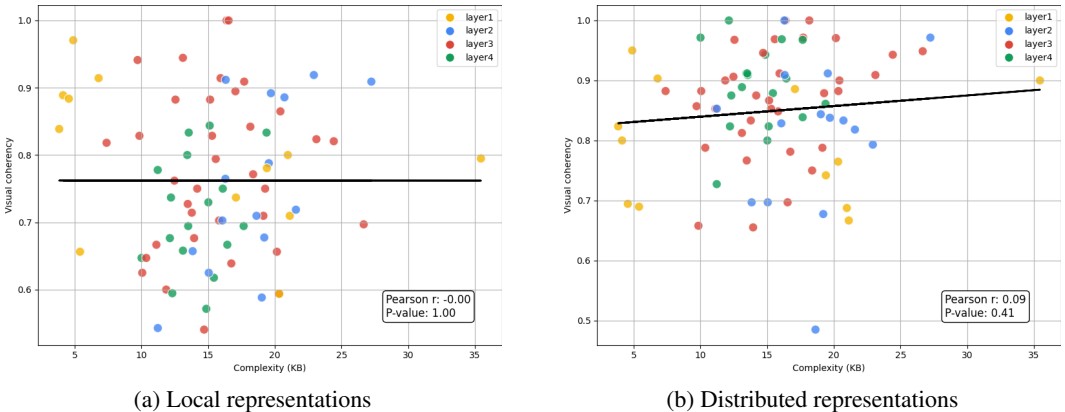

(a) Local representations                    (b) Distributed representations

Figure 13: **Kolmogorov complexity of features from a ResNet50 (Exp II).**

## G.2  Similarity

**Structural similarity Index**  The second hypothesis we test is if the images used to illustrated distributed features are more similar to the ones used to illustrate local features. The intuition behind this hypothesis is that the more ambiguous the features, the more dissimilar the images used to illustrate it, the worse people will perform in our experiment. To test this hypothesis we used an existing similarity measure, namely the structural similarity index measure (Wang et al., 2004; Wang & Bovik, 2009). This measure quantifies the similarity of 2 images based on luminance, contrast and change in structural information.

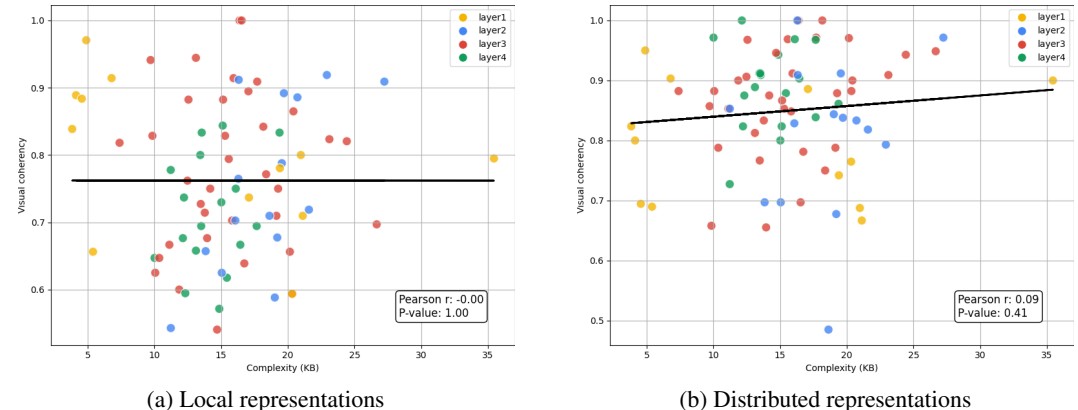

(a) Local representations        (b) Distributed representations

Figure 14: **Kolmogorov complexity of features from a VGG16 (Exp IV).**

**Methodology.** For a given target feature, we have access to T ∈ [1, 10], based on how many trials can be semantically controlled). In each trial, we have 10 images that possess the target feature. We measure the structural similarity index across every possible pair of these 10 images and calculate the average SSIM for that trial. We apply this methodology for every trial of a given feature and then average the SSIM values across all trials to obtain a measure of similarity for that feature. This process is applied to every feature.

**Results.** In both cases, we do not find a significant correlation between the structural similarity Index and the visual coherency scores from both Experiments II (ResNet50) (Fig. 15) and IV (VGG16) (Fig 16) suggesting that the visual coherency is driven by more complex factors that only luminance, constrast and the structure of the images.

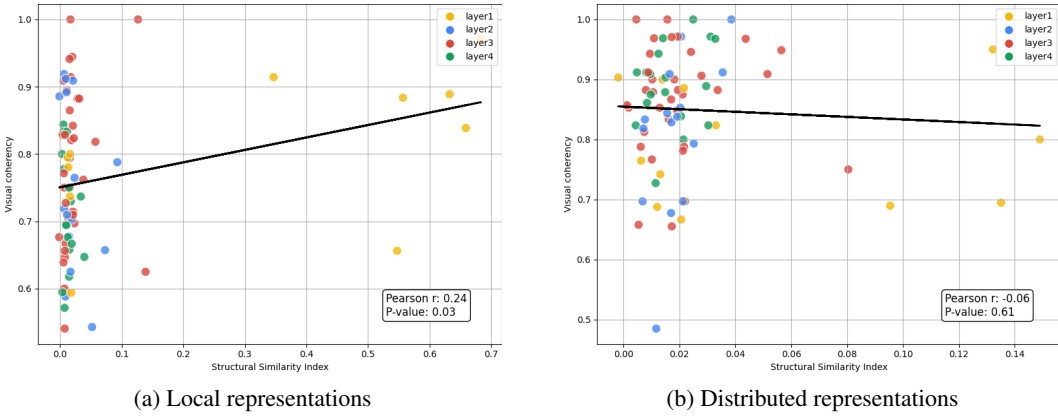

(a) Local representations        (b) Distributed representations

Figure 15: **Structural Similarity of images used to illustrate features from a ResNet50 (Exp II).**

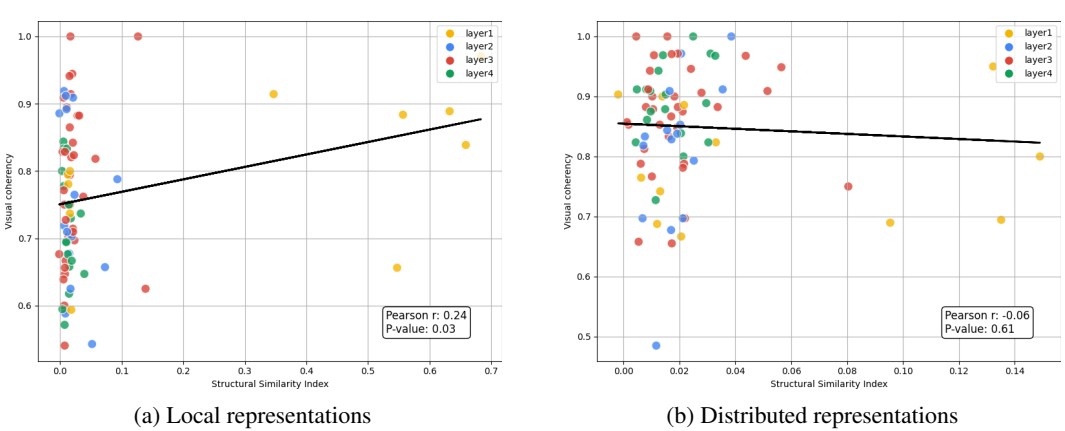

(a) Local representations          (b) Distributed representations

Figure 16: **Structural Similarity of images used to illustrate features from a VGG16 (Exp IV).**

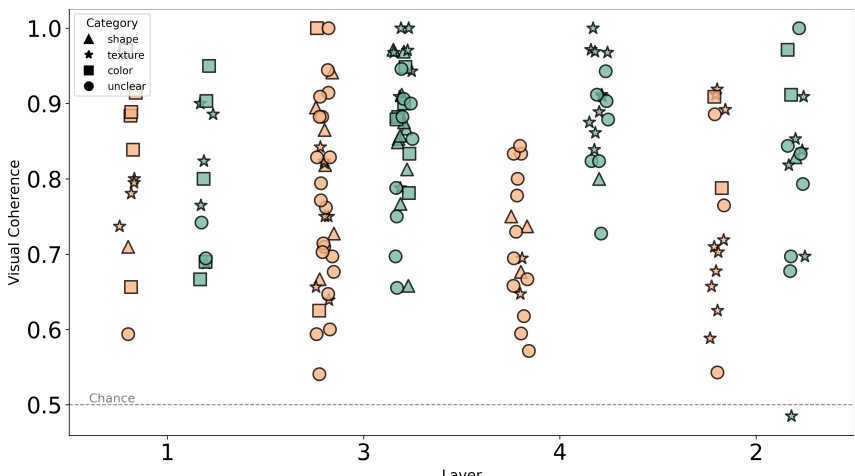

Figure 17: **What types of features are more or less interpretable.**

## H  WHAT TYPE OF FEATURES ARE MORE INTERPRETABLE?

To address the reviewer's suggestion, we attempt to provide intuition about which types of features are more or less interpretable by manually categorizing them into four categories: color, shape, texture, and unclear. For the definitions of shape and texture, we follow those from Gatys et al. (2017) and Geirhos et al. (2018). They define shape as "the set of contours that describe the 3D form of an object," and texture as "an image (region) with spatially stationary statistics. Note that on a very local level, textures (according to this definition) can have non-stationary elements (such as a local shape): e.g., a single bottle clearly has non-stationary statistics, but many bottles next to each other are perceived as a texture: 'things' become 'stuff'." The results are presented in Figure 17.

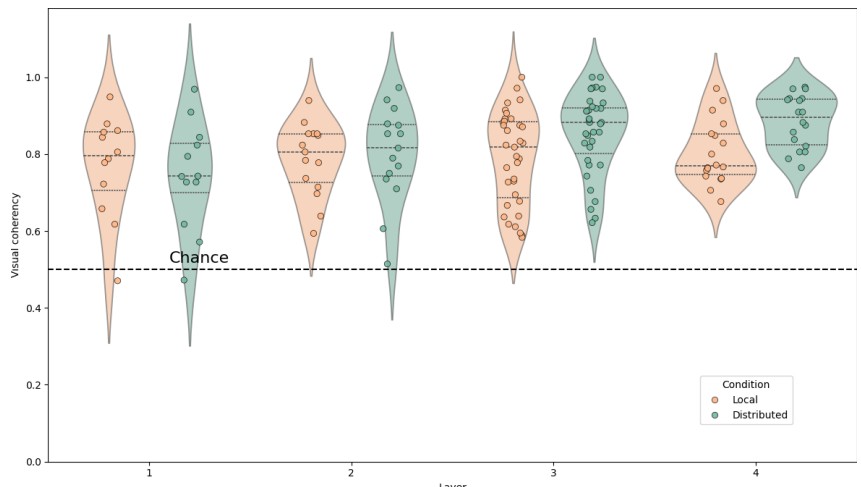

Figure 18: **Per-layer results for Experiment I**

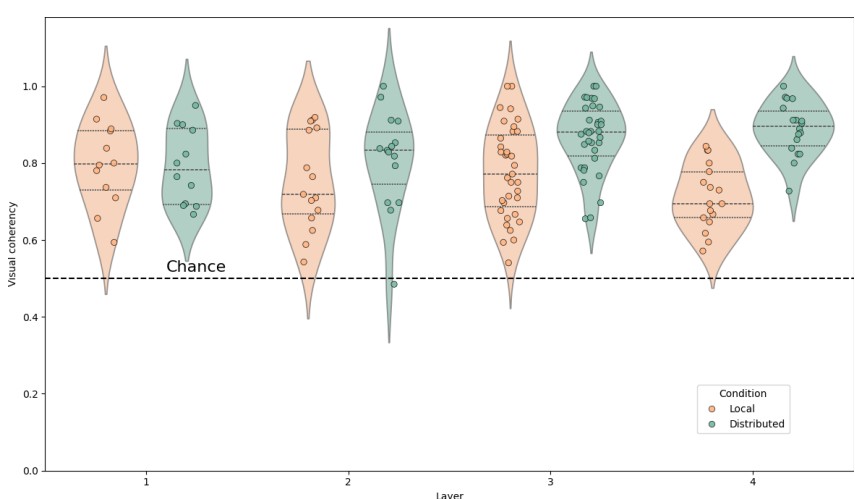

Figure 19: **Per-layer results for Experiment II**

# I ADDITIONAL PLOTS

As suggested by a reviewer, we replot in Fig 18-19-20 the results from Experiment I, II, and III in a way that can foster a more truthful appreciation of the effects.

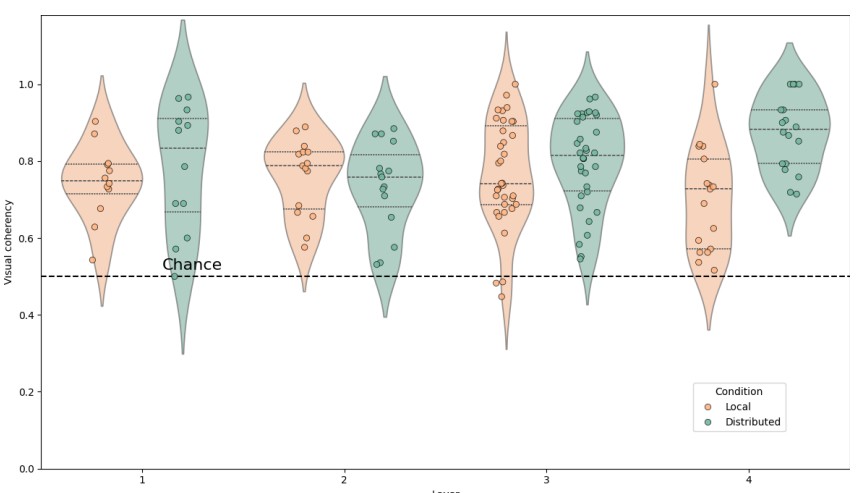

Figure 20: **Per-layer results for Experiment III**

