# OpenReview forum: "Local vs distributed representations: What is the right basis for interpretability?"
_ICLR.cc/2025/Conference — Submitted to ICLR 2025_

### Official Review · Reviewer_ZM3C · 2024-11-01

**Soundness:** 2
**Presentation:** 3
**Contribution:** 3
**Rating:** 6
**Confidence:** 4

**Summary:**

The authors perform computational and psychophysics experiments to determine whether local representations are more interpretable and useful for model decisions than distributed representations. They conclude that distributed representations are more interpretable and models rely more on them.

**Strengths:**

* Well written
* Well referenced
* Addresses an important issue for the interpretability community
* Provides evidence of interpretability from human perception
* Provides evidence of feature importance for model decisions

I find it hard to comment on originality (e.g., the work is important/useful regardless of whether the methods themselves are novel). The 4th bullet above is to me an important component which is not always assessed rigorously in other work.

**Weaknesses:**

I provide a summary first and then elaborate.

* The paper makes general claims about distributed vs local representations but does not provide empirical evidence for this generality or theoretical arguments for why it should be expected.
* Does not make explicit the model of human cognition/perception that motivates the psychophysics experiments and their analyses.
* Does not provide theoretical insight for why sparse distributed representations obtained via NMF should match human-interpretable features.
* Dichotomizes local vs distributed in the interpretation of the results, even though the results indicate the difference might be one of degree.
* Properties of the sampling of neurons are unclear.

Line 093: “…growing consensus that distributed representations constitute a stronger basis for interpretability compared to local ones...” This is the main point of the introduction and should be elaborated and explained. It merits more space than the generic survey-like paragraphs that come before it. In particular, what is the theoretical rationale for why this should be the case? Or is this ‘consensus’ purely driven by empirical findings? In any case, it would be helpful to clarify.

Line 084: “representations driven by single features are more desirable because they are expected to be easier to understand by a human” Why is this so? Single features can be non-intelligible to humans. Otherwise the definition of feature would have human-intelligibility baked in, in which case the quoted statement is circular.

Line 257: “Measure the ambiguity, or perplexity, of the visual feature as a proxy for its interpretability.” But a neuron can be interpretable even though it is ‘ambiguous’ (e.g., because it responds to more than one feature). For instance, it seems to me that responding to a disjunction of features is an interpretable response.

Perhaps these issues can be clarified if the authors make explicit a (ideally formal) notion of human-intelligible feature.

The model of human perception is left implicit in the manuscript. What is the model of human cognition that motivates the predicition that coherence relates to interpretability?

Line 275 mentions “Representative sample of units”. What are they representative of? And how is the representativeness achieved? Relatedly, can the authors clarify why they select the 80 units reported in Zimmerman? How was this selection made?

Do the results generalize beyond ResNet50 and ImageNet? Do they generalize across modalities (vision, audition, language, etc)? As far as I can see, there is no empirical evidence of generalization in these directions in the paper, nor theoretical arguments why it should be expected. Nevertheless, the paper makes a general claim about local versus distributed representations. It would be helpful to have experiments testing generalization in these directions and/or rigorous theoretical arguments.

Figure 1: As with other figures in the manuscript, it might benefit from unambiguously pointing to figure elements in the text. For instance, “depicted in the images at the bottom” should presumably point only to the visual elements in orange, not to all visual elements at the bottom of panel (a). Furthermore, panel (a) contains multiple other elements in the graphs which are not discussed in the figure caption (e.g., axes, black and colored data points, “n_1” labels, etc.). It would be great if the authors can take the reader through each panel explaining how the schematic depicts the concepts the authors want us to take away. Additional issues include: orange and green colors are referenced in the figure caption but gray is not. Likewise, the explanation in Figure 4 could be clearer. For example, you can clearly identify each experiment and avoid ambiguoous referents such as “this set of images”.

Participants perform quite well in the local representation condition. This means, following the authors’ rationale, that the local representation is quite interpretable as well. However, the authors express a more dichotomic view throughout. Why is this so? It seems to me that a dichotomic interpretation of the results is not warranted. At the very least, consideration of mechanisms that use both local and distributed representations, and what each of these contribute to, is called for.

**Questions:**

Minor comments, suggestions and questions:

Line 214 reads “neuron in \mathcal{A}”. Shouldn’t this be a component of a vector $a \in \mathcal{A}”?

Line 213: why is v_c indexed by c if it is made to correspond to an image x_i indexed by i?

Line 215: notation for modified activations a’_i is using a font different from the one in the definition in the previous paragraph.

Line 242: “by measuring as the difference” should presumably be “by measuring the difference”

The parallel between the problems of XAI and neuroscience, highlighted in the second and third paragraphs of the introduction, has been drawn explicitly and elaborated on by Vilas et al (2024) in the context of interpretability.

Please clarify mathematical notation. For instance, in equation 1, is y_i a vector? How does equation 1 relate to importance?

Stimuli selection: is it possible that the selection of maximally and minimally activating images contribute to mischaracterization of the full neural function? Relatedly, why are there more strongly activating images than weakly activating images?

Could the authors plot/overlay the data points in the graphs (i.e., not just bars)? It could foster a more truthful appreciation of the effects.

Line 460 references figure 7, which appears in the appendix, as if it were in the main text. There are possibly other instances of this throughout.

Given the authors' conclusions favoring distributed representations, the results could be relevant to the following literature exploring feasible interpretability queries, most of which only explore axis-aligned queries: Adolfi et al., 2024; Barcelo et al., 2020.

It might be useful to relate local versus distributed representations to discussions of axis-aligned versus non-aligned features.

Barceló, P., Monet, M., Pérez, J., & Subercaseaux, B. (2020). Model Interpretability through the lens of Computational Complexity. Advances in Neural Information Processing Systems, 33, 15487–15498. https://proceedings.nips.cc/paper_files/paper/2020/hash/b1adda14824f50ef24ff1c05bb66faf3-Abstract.html

Adolfi, F., Vilas, M. G., & Wareham, T. (2024). The Computational Complexity of Circuit Discovery for Inner Interpretability (No. arXiv:2410.08025). arXiv. http://arxiv.org/abs/2410.08025

Vilas, M. G., Adolfi, F., Poeppel, D., & Roig, G. (2024). Position: An Inner Interpretability Framework for AI Inspired by Lessons from Cognitive Neuroscience. Proceedings of the 41st International Conference on Machine Learning, 49506–49522. https://proceedings.mlr.press/v235/vilas24a.html

---

> ### Author Response · Authors · 2024-11-26
>
> We thank the reviewer for the very thorough and very interesting discussions which will allow us to significantly improve the clarity of the paper.
>
>
> # Concerns
>
> # Clarification about Line 093: “…growing consensus that distributed representations constitute a stronger basis for interpretability compared to local ones...”
>
> Thank you for the feedback; we will clarify the manuscript – the consensus is driven by empirical findings in large language models (Bricken et al. 2023 & Templeton et al. 2024 & Rajamanoharan et al. 2024) and theoretical results (Elhage et al. 2023 & Scherlis et al. 2023) with small datasets suggesting inherent limitations of local representations, but evidence in the field of computer vision are lacking, which is the gap our work aims to fill. We will clarify this point in the final version of the manuscript.
>
> Bricken, et al., "Towards Monosemanticity: Decomposing Language Models With Dictionary Learning", Transformer Circuits Thread, 2023.
>
> Templeton, et al., "Scaling Monosemanticity: Extracting Interpretable Features from Claude 3 Sonnet", Transformer Circuits Thread, 2024.
> Rajamanoharan, Senthooran, Tom Lieberum, Nicolas Sonnerat, Arthur Conmy, Vikrant Varma, János Kramár, and Neel Nanda. "Jumping ahead: Improving reconstruction fidelity with jumprelu sparse autoencoders." arXiv: 2024.
>
> Elhage, N., Hume, T., Olsson, C., Schiefer, N., Henighan, T., Kravec, S., Hatfield-Dodds, Z., Lasenby, R., Drain, D., Chen, C. and Grosse, R., 2022. Toy Models of Superposition. Transformer Circuits Thread, 2022
>
> Scherlis, Adam, Kshitij Sachan, Adam S. Jermyn, Joe Benton, and Buck Shlegeris. "Polysemanticity and capacity in neural networks." arXiv 2023.
>
> ## L.84 & L.257: Why do we assume that single features are expected to be easier to understand? Can ambiguous neurons be interpretable?
>
> We are making the hypothesis that single features are expected to be easier to understand based on previous works—visual features (Olah et al. 2020, Goh et al. 2021) but also more broadly features as coherent elements in NLP (Bricken, et al. 2023, Templeton et al. 2024). The assumption is that the more ambiguous the neuron the harder it might be to interpret, but we do not claim that interpretability is impossible if a neuron is ambiguous, just that it might take (significantly) more effort to decipher the more complex firing pattern. We will clarify this point in the revised version of the manuscript.
>
> Olah, et al., "Zoom In: An Introduction to Circuits", Distill, 2020.
>
> Goh, et al., "Multimodal Neurons in Artificial Neural Networks", Distill, 2021.
>
> Bricken, et al., "Towards Monosemanticity: Decomposing Language Models With Dictionary Learning", Transformer Circuits Thread, 2023.
>
> Templeton, et al., "Scaling Monosemanticity: Extracting Interpretable Features from Claude 3 Sonnet", Transformer Circuits Thread, 2024.
>
>
> ## The model of human perception is left implicit in the manuscript. What is the model of human cognition that motivates the predicition that coherence relates to interpretability?
>
> Interpretability means identifying exactly the features learned by the model. This is usually done by looking at a set of images, where all images possess the features. An image can be thought of as a collection of visual elements and the identification of a common visual element across a set of images can be thought of as a matching process across all images—all visual elements (Tversky 1977). As the number of perceptually different visual elements increases (which would be the case for ambiguous neurons), the visual complexity of the task increases (Olivia et al. 2004), hence identifying the correct features can be expected to become harder. We will clarify this point in the final version of the manuscript.
>
> Tversky, A. (1977). Features of similarity. Psychological review, 84(4), 327.
>
> Olivia, A., Mack, M. L., Shrestha, M., & Peeper, A. (2004). Identifying the perceptual dimensions of visual complexity of scenes. In Proceedings of the annual meeting of the cognitive science society (Vol. 26, No. 26).

---

> > ### Author Response · Authors · 2024-11-26
> > **Official Comment by Authors pt.2**
> >
> > ## Perhaps these issues can be clarified if the authors make explicit a (ideally formal) notion of human-intelligible feature.
> >
> > We believe that a feature is human-intelligible when people can easily identify what visual element the feature is about. In the case of our psychophysics experiment, it means that given a set of images that possess the visual elements, participants easily identify this visual element, and they use this to find the query image that also possesses the visual element.
> > A formal definition of what makes a feature intelligible would require a perfect model of our ability to process visual information. Nevertheless, in general, we believe that human-intelligibility is impacted by many factors, mainly the number of visual elements encapsulated by the feature as well as the dissimilarity of those visual elements, both together can be thought of as a notion of complexity to the intelligibility. When testing hypotheses to identify the feature, we are limited by cognitive constraints linked to the number of elements we can manipulate, this number is impacted by how perceptually different the visual elements are.
> >
> >
> > ## Properties of the sampling of neurons are unclear
> >
> > We would like to redirect the reviewer to the global response for a clarification.
> >
> >
> > ## Generalization of our results.
> >
> >  We would like to redirect the reviewer to the global response for our response to this concern.
> >
> >
> > ## Dichotomy in the interpretation of the results.
> >
> > This is an interesting point that we hope to clarify in the final version of the manuscript. Yes, participants can perform reasonably well in the local conditions. We claim that distributed representations are broadly superior because participants never seem to perform significantly worse using them. Hence, while the results suggest that distributed representations are not always necessary, there appears to be no downside to studying features using them. That being said, it is true that it is still not clear why they are not always a necessity and when they are. We leave this open question for future works.
> >
> > ## Figures’ clarity.
> >
> > Thank you for the helpful feedback, we will make the caption of Figure 1 & 4 clearer in the final version of the manuscript.
> >
> > ## Addressing the typos and clarifying mathematical notations
> >
> > L.213: i is regarding the index of the images in the set of images used to trained CRAFT (e.g., i ∈ [0 , 299]), while c is the index of the vector in the representations (e.g., c ∈ [0 , 2047]).
> > L.214, 215, 242: Yes, thank you for noticing the typo.
> >
> > We will correct those typos and clarify those points in the manuscript.
> >
> >
> > ## How does equation 1 relate to importance?
> >
> > Equation 1 measures the drop in the probability of the original decision when nulling the dimension of interest (neuron or distributed vector). Measuring the impact of removing an input dimension on the output of a function is standard practice when evaluating explainability methods. The aim of this sensitivity analysis is to evaluate how well explainability methods find the most important pixels (Petsiuk et al. 2018) or latent space dimensions (Fel et al. 2023) to explain a decision.
> >
> > Petsiuk, Vitali, Abir Das, and Kate Saenko. "RISE: Randomized Input Sampling for Explanation of Black-box Models." BMVC 2018.
> >
> > Fel, Thomas, Victor Boutin, Louis Béthune, Rémi Cadène, Mazda Moayeri, Léo Andéol, Mathieu Chalvidal, and Thomas Serre. "A holistic approach to unifying automatic concept extraction and concept importance estimation." NeurIPS 2024.
> >
> > ## Stimuli selection: is it possible that the selection of maximally and minimally activating images contribute to mischaracterization of the full neural function? Relatedly, why are there more strongly activating images than weakly activating images?
> >
> > Could the reviewer please clarify why they believe that our selection of images could have contributed to the mischaracterization of the full neural function?
> > The only mischaracterization we are aware of is the intrinsic limitation of maximally and minimally activating images which do not offer a complete view of the full neural function.
> > The choice for the cut-off at 2500 is based on (Zimmerman et al. 2023; see Fig 6), as they show that participants can perform the task well using images up to top 2500. The cut-off of 400 is arbitrary/computational.
> >
> >
> > ## Could the authors plot/overlay the data points in the graphs (i.e., not just bars)? It could foster a more truthful appreciation of the effects.
> >
> > As suggested by the reviewer, we have added another version of the graphs in the appendix I.
> >
> >
> > ## Literature recommendations
> >
> > Thank you, we were not aware of those papers. We will cite Vilas et al (2024), and we will read (Adolfi et al., 2024; Barcelo et al., 2020.) and include them in the related work.

---

> > > ### Comment · Reviewer_ZM3C · 2024-11-26
> > > **Response to rebuttal**
> > >
> > > On the request for clarification: my concern with mischaracterisation of function is precisely the one alluded to by the authors.
> > >
> > > For most of the critical issues I’ve raised, they have been approached by pointing to previous work (note however that precedent is not necessarily an indicator of soundness). The author’s implicit model of human cognition for the task, the idea of intelligible feature (quite circular as stated right now), the properties of the sampling of neurons (note: not merely the quantity), and the motivation for interpreting the results dichotomously (given that participants perform well with either representation), all of these aspects of the work remain underspecified or poorly motivated.
> > >
> > > I don’t think this necessarily undermines the work that is already there (that can only be answered when the above is made explicit), and I stand by the strengths I pointed out in my initial assessment. However, I consider the issues above substantial.
> > > Therefore, I thank the authors for the pointers and clarifications, and will keep the score.

---

> > > > ### Author Response · Authors · 2024-11-29
> > > >
> > > > We are grateful to the reviewer for engaging in the rebuttal, and we are eager to clarify ourselves further.
> > > >
> > > > ## Motivation for interpreting the results dichotomously
> > > >  We would like to redirect the reviewer to the global response for our response to this concern.
> > > >
> > > >
> > > > ## Mischaracterisation of function
> > > > Thank you for the clarification, we will add a discussion about the limitation of studying features through the lens of maximal images.
> > > >
> > > > ## Properties of the sampling of neurons
> > > > The motivation for reusing the same neurons is not only for simplicity and reproducibility   but also because we believe the sampling proposed in Zimmermann et al. 2023 to be sound. While the quantity has been justified with a priori analysis based on early results, the sampling has been done fairly—without bias for certain layers—to get a representative sample.
> > > >
> > > > *“For each of the investigated models, we randomly select [...] units by first drawing a network layer from a uniform distribution over the layers of interest and then selecting a unit, again at random, from the chosen layer. This scheme is used instead of randomly drawing units from a uniform distribution over all units since CNNs typically have more units in later layers.”*
> > > >
> > > > We will revise the manuscript with a clearer motivation and we will add the details of the sampling methodology.
> > > >
> > > > ## The author’s implicit model of human cognition for the task
> > > > We are eager to clarify the motivations behind our experimental protocol, but first we would very much appreciate it if the reviewer would clarify what they mean by *“model of human cognition”*—the level or scope of it—, as this wording can be understood very broadly. As it is expressed, it is hard for us to respond satisfactorily to it.
> > > >
> > > > ## The idea of intelligible feature
> > > > We understand the concern of the reviewer, but given our current understanding of human visual perception, it is not possible to define formally what an intelligible feature is. Instead, we default to a functional definition: a feature is intelligible if people can match images that share this feature.

---

### Official Review · Reviewer_VYZn · 2024-11-01

**Soundness:** 3
**Presentation:** 3
**Contribution:** 2
**Rating:** 5
**Confidence:** 4

**Summary:**

In this manuscript, the authors present the results of 3 psychophysics experiments on the interpretability of internal representations of a ResNet-50 trained on ImageNet. Following an established paradigm the authors showed images that highly activate a dimension and images that lowly activate a dimension in the deep neural network and asked subjects to infer whether a new test image activates the dimension or not. The main comparison is between the original neural dimensions and distributed directions determined by CRAFT, a dictionary learning method. The authors find better performance for the distributed representations. Nulling the most activated distributed direction also impaired network performance more than nulling the most active neuron. All effects get stronger for deeper layers in the network.

**Strengths:**

In general, interpretability of neural networks is a pressing matter and the sparse bases for interpretability are a recent development that is worth testing. To do so, the authors present a coherent set of experiments with state of the art methods. Both in terms of the psychophysical task and in terms of the statistical analysis, I believe the results are solid.

**Weaknesses:**

While the paper seems sound enough to me, the I doubt the importance of these results. It appears to be a solid confirmation of ideas we had before with established methods. This is good to do, but of somewhat limited impact.

- The results presented here are limited to a single network for a single network for a single task. Thus it remains unclear whether the effects described are general.
- All the technical steps were established before, i.e. there is no technical innovation in this paper
- As sparse dictionaries were developed as an XAI method, it is already a general believe in the literature that they are indeed more interpretable than the original neural activities, especially for deeper representations
- The effect on interpretability appears to be quite gradual, i.e. while the difference is clearly significant, the features extracted by CRAFT are still far from optimally interpretable especially for earlier layers. Thus, the problem of interpreting DNN activations remains unsolved effectively.
- There could be somewhat deeper analysis of the data, for example which features are more or less interpretable, what proportions of features are interpretable, etc.

**Questions:**

Can the results be used to improve interpretability somehow?

Can the results be reused for other networks in some way?

---

> ### Author Response · Authors · 2024-11-26
>
> We thank the reviewer for the helpful feedback. We address the reviewer’s concerns below.
>
> # Concerns
>
> ## Doubt about the importance of results.
>
> As the reviewer correctly stated, there is a "general belief" that sparse dictionaries are better—and most of this comes from NLP, not vision (Bricken, et al. 2023, Templeton et al. 2024). Our contribution is to experimentally verify this belief with empirical evidence. We will better highlight this contribution in the final version of the manuscript.
>
> Bricken, et al., "Towards Monosemanticity: Decomposing Language Models With Dictionary Learning", Transformer Circuits Thread, 2023.
>
> Templeton, et al., "Scaling Monosemanticity: Extracting Interpretable Features from Claude 3 Sonnet", Transformer Circuits Thread, 2024.
>
> ## Generalization of the results.
>
> We would like to point the reviewer to our general response.
>
>
> ## All the technical steps were established before, i.e. there is no technical innovation in this paper
>
> As stated by reviewer ZM3C “the work is important/useful regardless of whether the methods themselves are novel”.  This paper aims to evaluate the progress brought by current approaches for studying distributed representations. Our focus is on collecting human insights to inform future methods rather than improving existing ones.
>
>
> ## Features extracted by CRAFT are still far from optimally interpretable especially for earlier layers. Thus, the problem of interpreting DNN activations remains unsolved effectively.
>
> We do not claim distributed representations solved interpretability. We claim that distributed representations offer a more promising basis moving forward as they appear to have no downside compared to alternatives, and we verify this claim with experimental evidence. Unless we misunderstood the concern of the reviewer, we fail to see why the non-perfect score of distributed representations is a weakness of the paper.
>
>
> ## Deeper analysis of the data.
>
> Thank you for the interesting suggestion. We have done a tentative categorization of the interpretable features, which can be found in appendix H.
>
>
> # Questions
>
> ## Can the results be used to improve interpretability somehow?
>
> Can the reviewer clarify what they mean exactly by “improve interpretability”?  Our paper is focused on validating which basis is more promising for interpretability, and the results suggest that further improvements in the interpretability of DNNs’ features will be achieved through a focus on distributed representations, be it creating novel methods to find better-distributed representations or explaining them. Additionally, our result might hint at the benefit of sparse representations, hence imposing sparsity constraints on models during training might lead to more interpretable models.
>
> ## Can the results be reused for other networks in some way?
>
> We believe that this question is addressed in our global response about the generalizability of our results. If this is not the case, could the reviewer please clarify the aim of this question, thank you.

---

> > ### Comment · Reviewer_VYZn · 2024-11-26
> > **Acknowledgement of receipt**
> >
> > I wanted to acknowledge the authors responses. It is great to see that the authors are adding comparisons for another network and generally work on strengthening their results. Nonetheless, I don't really see how these results move us substantially towards more interpretable DNNs.
> >
> > The authors describe their results as if they decisively showed that distributed representations are the right way to go. The difference between the two specific methods is only a few percent in human recognition though. My interpretations of this is: Both methods are still pretty bad, currently the CRAFT ones do slightly better. For the question, how we might find actually good methods in the future, I think these results are a vague hint at best.
> >
> > The ultimate goal of finding better help for interpreting DNNs is also the motivation for my later questions that the authors essentially answer negatively: Some other psychophysical data like performance on distorted images can be used to evaluate new methods and DNNs. This fact makes the data more valuable for the field than the ones collected for this manuscript. Similarly, getting any more concrete hints on how future interpretation methods could be constructed would make these results more valuable. And finally, new methods for evaluation would be a contribution, too.
> >
> > Based on this assessment, I am going to keep my scores, as I still view them as fitting.

---

> > > ### Author Response · Authors · 2024-11-29
> > >
> > > We thank the reviewer for taking the time to read our response. We would like to redirect the reviewer to our general comment for our response to the concern regarding the interpretation of our results.

---

### Official Review · Reviewer_QXBx · 2024-11-04

**Soundness:** 3
**Presentation:** 3
**Contribution:** 3
**Rating:** 6
**Confidence:** 4

**Summary:**

This paper studies the question of human interpretability of the features derived from deep neural network. They considered two types of features: (i) features that triggered maximal responses (maximally activating images) in certain network units; (ii) features that were obtained by applying sparse dictionary learning to the maximally activating images. They defined these two classes as local representation v.s. distributed representation, respectively. Using psychophysics experiments, the authors studied which type of features were more interpretable. They reported some evidence suggesting that the sparse features was more interpretable to human observers. Overall, this seems to be an interesting piece of work, despite that the effect size obtained in the experiments is fairly small.

**Strengths:**

Originality: While this is not the first study to use psychophysics to understand the interpretability of deep network features, human interpretability of deep network features remains an under-explored direction. Overall, the combined experimental and modeling approach is considered to be novel.


Quality: The study is generally well-executed. Behavioral responses were collected from a large number of human subjects.  Three experiments were conducted. The first one was the main experiment. The remaining two were control experiments to help further interpreting the results.


Clarity: The method and approach are generally well described, except for a few sections (see comments below about Section 4).


Significance: Understanding and interpolating the features of deep nets is an interesting and timely question. The approach used here, while not entirely novel, remains useful.

**Weaknesses:**

- The formulation of local v.s. distributed representation (two concepts that are essential to this paper) is messy and problematic in the current version. In the Introduction, the author referred to the grandmother cell as the local representation, and cited the paper by Quiroga et al (2005). I think the results in Quiroga et al (2005) was misinterpreted. The fact that, from a small of recorded neurons in the hippocampus in Quiroga et al (2005) one could already find a neuron that selective to a specific person, suggests that there is a large population of neurons that are selective to that person (yet being sparse). Thus, the results from Quiroga et al (2005) as well as other related studies instead favor a sparse and distributed representation.  I wonder if a better distinction would be “dense” v.s. “sparse”, rather than “local” v.s. “distributed”.

Thus, I think the overall pitch of the paper, i.e., the formulation of “local” v.s. “distributed”, is problematic.


- One potential explanation for the main experimental results might be that the complexity of the features from the two classes (original maximally activating images v.s. sparse features obtained from them) did not match. Perhaps the sparse features used in the experiments are simpler and have less entropy compared to the maximally activating images. Would it be able to rule this out?

- The work only tested one version of ResNet50. It remains unclear whether the results would generalize to other deep nets. I understand that it would be too much to re-run the experiments using features from a different network for the current study. However, if the authors could compare the features in different networks (i.e., running these psychophysics in silico), that would likely strengthen the paper.

- The main effect, while being statistically significant, the effect size as reported in Section 4 was fairly small.  The paper needs to take the effect size into account when interpreting the results.

- While the number of subjects was large, the number of responses collected in each experiment (~5k) did not exceed the number in typical psychophysical experiments. Some of the statements regarding the scale of experiments should be toned down accordingly.


- The AIC comparison in line 450-451 implies a small difference (i.e, 4) between the two models. I think the interpretation should be more nuanced.

- How the Result section was presented can be improved. It would be useful if the authors could further unpack the meaning of the analyses performed there.

**Questions:**

What are the assumptions behind the logistic regression analysis in Section 4? It would be good to make these more explicitly so a naive reader would have a better sense of how the results should be interpreted.


Line 301-303, it was not exactly clear in terms of how the particular direction was selected. What does “the most frequently the most activated” mean? Please clarify.

Line 130: Re: “neurons’ tendency to respond to multiple unrelated visual elements”
What does “unrelated visual elements” mean? This seems to be fairly arbitrary.

Are the comparisons in Fig. 4 statistically significant? What do the error bars represent?

Line 277-281, what were special about the 80 neurons selected? Are these a representative subset of all units in the network? If not, how do we know if the results would generalize to other neurons in the network?

---

> ### Author Response · Authors · 2024-11-26
>
> Thank you for the interesting questions and the helpful suggestions for further experiments. We address both below.
>
> # Concerns
>
> ## Formulation of local vs distributed is problematic.
>
> > The formulation of local v.s. distributed representation (two concepts that are essential to this paper) is messy and problematic in the current version. In the Introduction, the author referred to the grandmother cell as the local representation, and cited the paper by Quiroga et al (2005). I think the results in Quiroga et al (2005) was misinterpreted. The fact that, from a small of recorded neurons in the hippocampus in Quiroga et al (2005) one could already find a neuron that selective to a specific person, suggests that there is a large population of neurons that are selective to that person (yet being sparse). Thus, the results from Quiroga et al (2005) as well as other related studies instead favor a sparse and distributed representation. I wonder if a better distinction would be “dense” v.s. “sparse”, rather than “local” v.s. “Distributed”.
>
> We believe our interpretation of the results in Quiroga et al. (2005) is consistent with the original interpretation by the authors. To quote them: “[the] subset of MTL cells is selectively activated by different views of individuals, landmarks, animals or objects. This is quite distinct from a completely distributed population code and suggests a sparse, explicit and invariant encoding of visual percepts in MTL”. Multiple cells selective for a specific object implies redundant rather than a distributed code. We want to emphasize that we simply cite (Quiroga et al. 2005) as a general motivation to draw the parallel between neuroscience and XAI regarding how both fields tackle the question of how information is encoded within a system. With that said, we agree that the nuance and connection behind local vs distributed—how neurons encode information—and dense vs sparse—how selective the neuron code is—in DNNs is not clear. In this work, we focus on exploring the former—how information is encoded—but it happens that current approaches also used to get distributed representations also impose sparsity constraints. We leave it to future works to explore the benefits for interpretability of the sparsity of distributed representations. We will update the manuscript to reflect the reviewer’s feedback.
>
>
> ## Do features from local vs distributed differ in obvious ways.
>
> Following the suggestion of the reviewer, we use existing metrics to probe if features from local vs. distributed representations differ in obvious ways. The first hypothesis we test is whether features from distributed representation are easier to interpret than features from local conditions because they are less complex. The second hypothesis we test is whether the images used to illustrate distributed features are more similar to the ones used to illustrate local features. The intuition behind this hypothesis is that the more ambiguous the features, the more dissimilar the images used to illustrate them, the worse people will perform in our experiment. In both cases, we did not find a correlation between either complexity or similarity with the performance of participants in our psychophysics experiments. Further details about those experiments can be found in Appendix G.
>
> ## Generalization of our results.
>
>  We would like to point the reviewer to our general response.
>
>
> ## Concerns about the effect size
>
> There seems to be a misunderstanding about which experiment constitutes the main results of this paper. The main results come from Experiment II, not from Experiment I, as the protocol used in II improves over the one used in I. Whereas the effect size is small in I, β = -0.47; p=0.04, it is medium in II, β=-0.93; p<0.001—respectively d=0.26 and d=0.51 when converted to Cohen’s d (Sánchez-Meca et al. 2003).
>
> Sánchez-Meca, Julio, Fulgencio Marín-Martínez, and Salvador Chacón-Moscoso. "Effect-size indices for dichotomized outcomes in meta-analysis." Psychological methods 8, no. 4 (2003): 448.avg_cosin
>
>
> ## Claims about the scale of experiment.
>
> Thank you for the feedback, we will take this comment into consideration and tone down the final version of the manuscript.
>
>
> ## Clarification about the AIC comparison
>
> We appreciate the reviewer’s comment. We will change the phrasing in the main text from “Based on the Akaike Information Criterion, the model without the interaction was preferred (9116 vs. 9120)” to a more careful “We fitted two GLMERs to our data: one including only the main effects of condition and semantic control, and another also considering the interaction between these two variables. Comparing these two models using the Akaike Information Criterion, we did not find evidence in support of an interaction (9116 vs. 9120 respectively, lower is better).”

---

> > ### Author Response · Authors · 2024-11-26
> > **Official Comment by Authors pt.2**
> >
> > # Questions
> >
> > ## Improve the presentation results section & What are the assumptions behind the logistic regression analysis in Section 4?
> >
> > We thank the reviewer for their comment and take their advice to heart.
> >
> > First, we will edit Section 4 to clarify how the logistic regression analyses are tied to our main hypothesis. Specifically, we operationalized interpretability as the probability for participants to select the correct query image in our behavioral task (following Borowski et al.’s protocol). We predict this probability will be lower in the local condition than in the distributed condition. We model this probability on a trial-by-trial level using a logistic regression with ConditionLocal as the main predictor. Defining the distributed condition as the base condition, we expect the regression coefficient for ConditionLocal to be negative.
> >
> > Second, to make the description of our regression analyses more explicit, we will add an appendix to the final version of the manuscript with equations describing the exact regression models we fitted. We will also include tables with the estimates for all their parameters. The equations will, for example, make it clear that a logistic regression assumes a linear relation between the predictors and the log odds of answering a trial correctly. We will also mention this assumption in the main text. The equations will show that we assume the random effects are normally distributed with a 0-mean.
> >
> > Finally, we will provide more clarification about the meaning of the random effects in the text. Accounting for these sources of variance in the data allows for better statistical inference and confirmatory hypothesis testing (Barr et al. 2013).
> >
> > Barr, Dale J., Roger Levy, Christoph Scheepers, and Harry J. Tily. "Random effects structure for confirmatory hypothesis testing: Keep it maximal." Journal of memory and language 68, no. 3 (2013): 255-278.
> >
> > ## Clarify selection direction l.301
> >
> > We select the dimension that is the most often the top1 activated across the 300 images. We will update the manuscript to clarify this.
> >
> >
> > ## What does “unrelated visual elements” mean?
> >
> > By unrelated, we mean a set of visual elements that do not seem to share anything visually. Defining formerly “unrelated visual elements” would be very hard, so we ground it on the empirical observations that a lot of researchers have come across neurons that respond to images that, according to their classifications, do not share common visual elements (Olah et al. 2020, Goh et al. 2021, Bricken, et al. 2023, Templeton et al. 2024).
> >
> > Olah, et al., "Zoom In: An Introduction to Circuits", Distill, 2020.
> >
> > Goh, et al., "Multimodal Neurons in Artificial Neural Networks", Distill, 2021.
> >
> > Bricken, et al., "Towards Monosemanticity: Decomposing Language Models With Dictionary Learning", Transformer Circuits Thread, 2023.
> >
> > Templeton, et al., "Scaling Monosemanticity: Extracting Interpretable Features from Claude 3 Sonnet", Transformer Circuits Thread, 2024.
> >
> >
> > ## Are the comparisons in Fig. 4 statistically significant? What do the error bars represent?
> >
> > The bars correspond to a 95% confidence interval. We did not make pairwise comparisons for layers, nor did we make claims about one. The aim of Fig.4 is to give an intuitive view of the results. Regardless, an indication of the significance of the difference emerges from the significant effect of depth we found in our GMLER analysis.
> >
> >
> > ## Why are the 80 neurons selected representatives of the ResNet50.
> >
> > We would like to redirect the reviewer to our general response.

---

> > ### Comment · Reviewer_QXBx · 2024-11-27
> >
> > Thank you for your response to the concerns I raised. The proposed revision will make the paper slightly better, although not substantially stronger as the observed effect is fairly small. I also largely agree with the concerns raised by other reviewers which have not been fully addressed. Overall, I feel this paper is a bit borderline, and am slightly more positive about it. So I will keep my original score.

---

> > > ### Author Response · Authors · 2024-11-29
> > >
> > > Thank you for taking the time to read our response. We would like to redirect the reviewer to our general comment for our response to the concern regarding the effect size of our results.

---

### Author Response · Authors · 2024-11-20
**Requesting further clarification to QXBx**

We want to thank all the reviewers for their valuable feedback. We have considered your comments and are currently conducting further experiments to address them in a general response. In the meantime, we just wanted to ask for clarification for a comment made by **QXBx**.




> One potential explanation for the main experimental results might be that the complexity of the features from the two classes (original maximally activating images v.s. sparse features obtained from them) did not match. Perhaps the sparse features used in the experiments are simpler and have less entropy compared to the maximally activating images. Would it be able to rule this out?

First, we want to clarify that both types of features, obtained from local or distributed representations, are illustrated in psychophysics experiments using maximally activating images. Then, we agree with the reviewer that exploring our results through the lens of feature complexity is very interesting, but in practice, what is the best way to measure the complexity of images is still an open research question. Hence, we would very much appreciate it if the reviewer could further clarify the kind of experiment that is suggested in the comment above. Thank you!

---

### Author Response · Authors · 2024-11-26
**General response**

We thank the reviewers for their time and effort and for providing constructive feedback that will help us significantly improve this work.

We are happy to read that reviewers considered that our paper is “well written” (ZM3C) and that our work addresses an “important” (ZM3C) and “timely [research] question” (QXBx) providing “evidence of interpretability from human perception” (ZM3C) through “coherent” (VYZn) and “well-executed” (QXBx) psychophysics studies.
In addition to individual responses to each reviewer’s questions or comments, we provide below a general response to some of the comments shared by multiple reviewers.




## Concerns about the generalizability of our results.

We believe that the reviewers’ concerns are warranted. At the same time, we would like to highlight that it is common practice in the interpretability literature involving the collection of human data to report results from single models (see e.g., Borowski et al. 2021, and Zimmermann et al. 2021). This being said, we have taken the reviewers’ comments to heart and have run additional experiments to improve the generalizability of our claims. First, regarding the concerns that our results result from evaluating a single model (QXBx & VYZn & ZM3C), we have conducted an additional psychophysics experiment on another DNN architecture where the results suggest that the superiority of distributed representation generalizes across architectures (see Appendix F for further details). Second, concerning the comment regarding modality (ZM3C), we will clarify in the final version of the manuscript that our general conclusion stems from our results in computer vision corroborating with the results obtained by others in NLP, suggesting that the effect generalizes across modalities. Third, we will discuss the impact of the dataset on our results (ZM3C).


## Why are the 80 neurons selected representatives of the ResNet50.

We selected the same neurons as those considered for the ResNet50 in Zimmerman et al. 2023 for simplicity. The motivation was articulated in the original study (Sect 3.1 and A.5). The authors chose the number of neurons based on a rigorous power analysis.

**We have uploaded an updated version of the manuscript, taking into account the reviewers' suggestions and the new results (Appendix F to I).**

Borowski, Judy, Roland S. Zimmermann, Judith Schepers, Robert Geirhos, Thomas SA Wallis, Matthias Bethge, and Wieland Brendel. "Exemplary Natural Images Explain CNN Activations Better than State-of-the-Art Feature Visualization." ICLR 2021.
Zimmermann, Roland S., Judy Borowski, Robert Geirhos, Matthias Bethge, Thomas Wallis, and Wieland Brendel. "How well do feature visualizations support causal understanding of CNN activations?." NeurIPS 2021.

---

### Author Response · Authors · 2024-11-29
**General comment 2**

We are grateful to the reviewers for engaging in the rebuttal and would like to take this opportunity to clarify the interpretation of our results.

While it remains an open question whether models encode features using a local code (individual neurons) or a distributed code (a population of neurons), our results suggest they use both. Indeed, participants perform well in the local condition but can benefit—sometimes significantly—from studying representations that do not enforce local encoding. This benefit may be small within a model but becomes important when comparing the interpretability of different models. **Our results illustrate this point well**: ResNet50 is as interpretable as VGG16 using local representations (76.2% ± 1.6 vs. 77.1% ± 1.6; z = -0.6, p = 0.4, Mann-Whitney U test) but becomes significantly more interpretable with distributed representations (85.1% ± 1.4 vs. 81.6% ± 1.5; z = -2.24, p < .001). Evaluating model interpretability solely through locally encoded features may unfairly penalize models that rely more on a distributed code.

**These new results are significant** because they challenge previous claims that scaling up models makes them less interpretable (Zimmermann et al. 2023). Our results suggest that their claim might no longer hold when considering distributed representations (although more work would be needed, which is beyond the scope of this paper).

Furthermore, we believe it important to put our results into context. While the community has been trying to make sense of local representations for close to two decades (Dumitru et al. 2009), studying a model’s representations through a new basis of vectors is a much more recent endeavor (Ghorbani et al. 2019) which only gained traction recently (Bricken et al. 2023). Therefore, as this area of research is still in its early stages, it is likely that future methods will continue to improve the interpretability of models.




Roland S Zimmermann, Thomas Klein, and Wieland Brendel. Scale alone does not improve
mechanistic interpretability in vision models. NeurIPS 2023.

Erhan, Dumitru, Yoshua Bengio, Aaron Courville, and Pascal Vincent. "Visualizing higher-layer features of a deep network." University of Montreal 1341, no. 3 (2009)

Amirata Ghorbani, James Wexler, James Y Zou, and Been Kim. Towards automatic concept-based explanations. NeurIPS 2019.

Bricken, et al., "Towards Monosemanticity: Decomposing Language Models With Dictionary Learning", Transformer Circuits Thread, 2023.

---

### Meta-Review · Area_Chair_4VDA · 2024-12-08

**Metareview:**

This paper examines the interpretability of distributed versus local representations in neural networks through psychophysical experiments with human participants. While the experimental methodology is sound and provides empirical evidence regarding interpretability of these two kinds of representations, the work has several critical limitations that preclude acceptance. The theoretical foundations explaining why distributed representations should enhance interpretability are inadequately developed. The generalizability of findings remains limited even with the addition of the VGG16 experiments, as results are confined to vision tasks and two specific architectures. In addition, the moderate effect sizes and above-chance performance with both representation types suggest a less dramatic difference between approaches than the paper's framing implies.

**Additional Comments On Reviewer Discussion:**

The final discussion centered on three key concerns: the significance of effect size in comparing distributed versus local representations, limited generalizability beyond the initial ResNet50 experiments, and insufficient theoretical foundations regarding interpretability enhancement. While the addition of VGG16 experiments partially addressed generalizability, reviewers noted that participants performed well with both representation types, suggesting the manuscript's framing overstates the advantages of distributed representations. The reviewers recommended developing a more explicit computational model and presenting findings more objectively rather than maintaining a stark dichotomy between representation types. I agree with these assessments.

---

### Decision · Program_Chairs · 2025-01-22

Reject